# ON THE IMPORTANCE OF CALIBRATION IN SEMI-SUPERVISED LEARNING

## ABSTRACT

State-of-the-art (SOTA) semi-supervised learning (SSL) methods have been highly successful in leveraging a mix of labeled and unlabeled data by combining techniques of consistency regularization and pseudo-labeling. During pseudo-labeling, the model's predictions on unlabeled data are used for training and thus, model calibration is important in mitigating confirmation bias. Yet, many SOTA methods are optimized for model performance, with little focus directed to improve model calibration. In this work, we empirically demonstrate that model calibration is strongly correlated with model performance and propose to improve calibration via approximate Bayesian techniques. We introduce a family of new SSL models that optimizes for calibration and demonstrate their effectiveness across standard vision benchmarks of CIFAR-10, CIFAR-100 and ImageNet, giving up to 16.2% improvement in test accuracy on the CIFAR-100-400-labels benchmark. Furthermore, we also demonstrate their effectiveness in additional realistic and challenging problems, such as class-imbalanced datasets and in photonics science.

## 1 INTRODUCTION

While deep learning has achieved unprecedented success in recent years, its reliance on vast amounts of labeled data remains a long standing challenge. Semi-supervised learning (SSL) aims to mitigate this by leveraging unlabeled samples in combination with a limited set of annotated data. In computer vision, two powerful techniques that have emerged are pseudo-labeling (also known as self-training) (Rosenberg et al., 2005; Xie et al., 2019b) and consistency regularization (Bachman et al., 2014; Sajjadi et al., 2016). Broadly, pseudo-labeling is the technique where artificial labels are assigned to unlabeled samples, which are then used to train the model. Consistency regularization enforces that random perturbations of the unlabeled inputs produce similar predictions. These two techniques are typically combined by minimizing the cross-entropy between pseudo-labels and predictions that are derived from differently augmented inputs, and have led to strong performances on vision benchmarks (Sohn et al., 2020; Assran et al., 2021).

Intuitively, given that pseudo-labels (i.e. the model's predictions for unlabeled data) are used to drive training objectives, the calibration of the model should be of paramount importance. Model calibration (Guo et al., 2017) is a measure of how a model's output truthfully quantifies its predictive uncertainty, i.e. it can be understood as the alignment between its prediction confidence and its ground-truth accuracy. In some SSL methods, the model's confidence is used as a selection metric (Lee, 2013; Sohn et al., 2020) to determine pseudo-label acceptance, further highlighting the need for proper confidence estimates. Even outside this family of methods, the use of cross-entropy minimization objectives common in SSL implies that models will naturally be driven to output high-confidence predictions (Grandvalet & Bengio, 2004). Having high-confidence predictions is highly desirable in SSL since we want the decision boundary to lie in low-density regions of the data manifold, i.e. away from labeled data points (Murphy, 2022). However, without proper calibration, a model would easily become over-confident. This is highly detrimental as the model would be encouraged to reinforce its mistakes, resulting in the phenomenon commonly known as *confirmation bias* (Arazo et al., 2019).

Despite the fundamental importance of calibration in SSL, many state-of-the-art (SOTA) methods have thus far been empirically driven and optimized for performance, with little focus on techniques that specifically target improving calibration to mitigate confirmation bias. In this work, we explore the generality of the importance of calibration in SSL by focusing on two broad families of SOTA

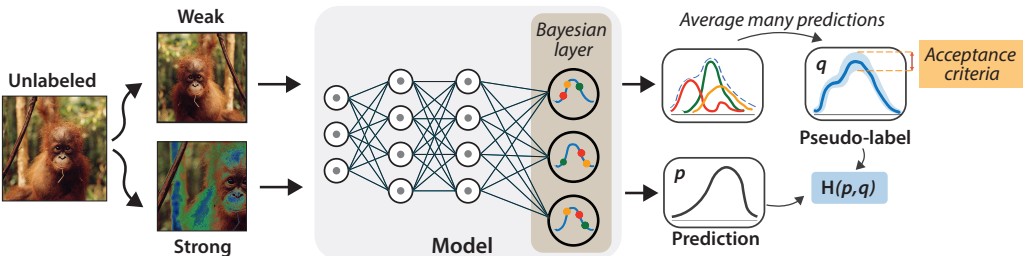

Figure 1: **Illustration of our BAyesian Model averaging (BAM) approach**

SSL methods that both use pseudo-labeling and consistency regularization: 1) threshold-mediated methods (Sohn et al., 2020; Xie et al., 2019a; Lee, 2013) where the model selectively accepts pseudo-labels whose confidence exceed a threshold and 2) "representation learning" methods adopted from self-supervised learning (Assran et al., 2021) where pseudo-labels are non-selective and training consists of two sequential stages of representation learning and fine-tuning.

To motivate our work, we first empirically show that strong baselines like FixMatch (Sohn et al., 2020) and PAWS (Assran et al., 2021), from each of the two families, employ a set of indirect techniques to *implicitly* maintain calibration and that achieving good calibration is strongly correlated to improved performance. Furthermore, we demonstrate that it is not straightforward to control calibration via such indirect techniques. To remedy this issue, we propose techniques that are directed to explicitly improve calibration by using approximate Bayesian techniques that are designed to capture model uncertainty. In our work, we explored approximate Bayesian neural networks (Blundell et al., 2015) and weight-ensembling approaches (Izmailov et al., 2018). Such Our modification forms a new family of SSL methods that improve upon the SOTA on both standard benchmarks and real-world applications. Our contributions are summarized as follows:

1. Using SOTA SSL methods as case studies, we empirically show that maintaining good calibration is strongly correlated to better model performance in SSL.

2. We propose to use approximate Bayesian techniques to directly improve calibration and provide theoretical results on generalization bounds for SSL to motivate our approach.

3. We introduce a new family of methods, BAM-, that improves calibration via BAyesian model averaging (see Fig. 1) and demonstrate their improvements upon a variety of SOTA SSL methods on standard benchmarks, notably giving up to 16.2% gains in test accuracy.

4. We further explored weight averaging techniques, one of which (i.e. EMA) being well-established in SSL and show that their effectiveness can be understood from improving pseudo-label calibration.

5. We further demonstrate the efficacy of BAM- in more challenging and realistic scenarios, such as class-imbalanced datasets and a real-world application in photonic science.

## 2 RELATED WORK

**Semi-supervised learning (SSL) and confirmation bias.** A fundamental problem in SSL methods based on pseudo-labeling (Rosenberg et al., 2005) is that of confirmation bias (Tarvainen & Valpola, 2017; Murphy, 2022), i.e. the phenomenon where a model overfits to incorrect pseudo-labels. Several strategies have emerged to tackle this problem; Guo et al. (2020) and Ren et al. (2020) looked into weighting unlabeled samples, Thulasidasan et al. (2019) and Arazo et al. (2019) proposes to use augmentation strategies like MixUp (Zhang et al., 2017), while Cascante-Bonilla et al. (2020) proposes to re-initialize the model before every iteration to overcome confirmation bias. Another popular technique is to impose a selection metric (Yarowsky, 1995) to retain only the highest quality pseudo-labels, commonly realized via a fixed threshold on the maximum class probability (Xie et al., 2019a; Sohn et al., 2020). Recent works have further extended such selection metrics to be based on dynamic thresholds, either in time (Xu et al., 2021) or class-wise (Zou et al., 2018; Zhang et al., 2021). Different from the above approaches, our work proposes to overcome confirmation bias in SSL by directly improving the calibration of the model through approximate Bayesian techniques.

**Model calibration and uncertainty quantification.** Proper estimation of a network's prediction uncertainty is of practical importance (Amodei et al., 2016) and has been widely studied. A common approach to improve uncertainty estimates is via Bayesian marginalization (Wilson & Izmailov, 2020), i.e. by weighting solutions by their posterior probabilities. Since exact Bayesian inference is computationally intractable for neural networks, a series of approximate Bayesian methods have emerged, such as variational methods (Graves, 2011; Blundell et al., 2015; Kingma et al., 2015), Hamiltonian methods (Springenberg et al., 2016) and Langevin diffusion methods (Welling & Teh, 2011). Other methods to achieve Bayesian marginalization also exist, such as deep ensembles (Lakshminarayanan et al., 2016) and efficient versions of them (Wen et al., 2020; Gal & Ghahramani, 2015), which have been empirically shown to improve uncertainty quantification. The concept of uncertainty and calibration are inherently related, where calibration is commonly interpreted as the frequentist notion of uncertainty. In our work, we will adopt some of these techniques specifically for the context of semi-supervised learning in order to improve model calibration during pseudo-labeling. While other methods for improving model calibration exists (Platt, 1999; Zadrozny & Elkan, 2002; Guo et al., 2017), these are most commonly achieved in a post-hoc manner using a held-out validation set; instead, we seek to improve calibration during training and with a scarce set of labels. Finally, in the intersection of SSL and calibration, Rizve et al. (2021) proposes to leverage uncertainty to select a better calibrated subset of pseudo-labels. Our work builds on a similar motivation, however, in addition to improving the selection metric with uncertainty estimates, we further seek to directly improve calibration via Bayesian marginalization (i.e. averaging predictions).

## 3 NOTATION AND BACKGROUND

Given a small amount of labeled data $\mathcal{L} = \{(x_l, y_l)\}_{l=1}^{N_l}$ (here, $y_l \in \{0,1\}^K$, are one-hot labels) and a large amount of unlabeled data $\mathcal{U} = \{x_u\}_{u=1}^{N_u}$, i.e. $N_u \gg N_l$, in SSL, we seek to perform a $K$-class classification task. Let $f(\cdot, \theta_f)$ be a backbone encoder (e.g. ResNet or WideResNet) with trainable parameters $\theta_f$, $h(\cdot, \theta_h)$ be a linear classification head, and $H$ denote the standard cross-entropy loss.

**Threshold-mediated methods.** Threshold-mediated methods such as Pseudo-Labels (Lee, 2013), UDA (Xie et al., 2019a) and FixMatch (Sohn et al., 2020) minimizes a cross-entropy loss on augmented copies of unlabeled samples whose confidence exceeds a pre-defined threshold. Let $\alpha_1$ and $\alpha_2$ denote two augmentation transformations and their corresponding network predictions for sample $x$ to be $q_1 = h \circ f(\alpha_1(x))$ and $q_2 = h \circ f(\alpha_2(x))$, the total loss on a batch of unlabeled data has the following form:

$$L_u = \frac{1}{\mu B} \sum_{u=1}^{\mu B} \mathbb{1}(\max(q_{1,u}) \geq \tau) H(\rho_t(q_{1,u}), q_{2,u}) \tag{1}$$

where $B$ denotes the batch-size of labeled examples, $\mu$ a scaling hyperparameter for the unlabeled batch-size, $\tau \in [0, 1]$ is a threshold parameter often set close to 1 and $\rho_t$ is either a sharpening operation on the pseudo-labels, i.e. $[\rho_t(q)]_k := [q]_k^{1/t} / \sum_{c=1}^{K} [q]_c^{1/t}$ or an $\mathrm{argmax}$ operation (i.e. $t \to 0$). $\rho_t$ also implicitly includes a "stop-gradient" operation, i.e. gradients are not back-propagated from predictions of pseudo-labels. $L_u$ is combined with the expected cross-entropy loss on labeled examples, $L_l = \frac{1}{B} \sum_{l=1}^{B} H(y_l, q_{1,l})$ to form the combined loss $L_l + \lambda L_u$, with hyperparameter $\lambda$. Differences between Pseudo-Labels, UDA and FixMatch are detailed in Appendix D.1.

**Representation learning based methods.** We use PAWS (Assran et al., 2021) as a canonical example for this family. A key difference from threshold-mediated methods is the lack of the parametric classifier $h$, which is replaced by a non-parametric soft-nearest neighbour classifier ($\pi_d$) based on a labeled support set $\{z_s\}_{s=1}^{B}$. Let $z_1 = f(\alpha_1(x))$ and $z_2 = f(\alpha_2(x))$ be the representations for the two views from the backbone encoder, their pseudo-labels $(q_1, q_2)$ and the unlabeled loss are given by:

$$q_i = \pi_d(z_i, \{z_s\}) = \sum_{s=1}^{B} \frac{d(z_i, z_s) \cdot y_s}{\sum_{s'=1}^{B} d(z_i, z_{s'})}; \quad L_u = \frac{1}{2\mu B} \sum_{u=1}^{\mu B} H(\rho_t(q_{1,u}), q_{2,u}) + H(\rho_t(q_{2,u}), q_{1,u}) \tag{2}$$

where $d(a, b) = \exp(a \cdot b / (\|a\| \|b\| \tau_p))$ is a similarity metric with temperature hyperparameter $\tau_p$ and all other symbols have the same meanings defined before. The combined loss is $L_u + L_{\text{me-max}}$

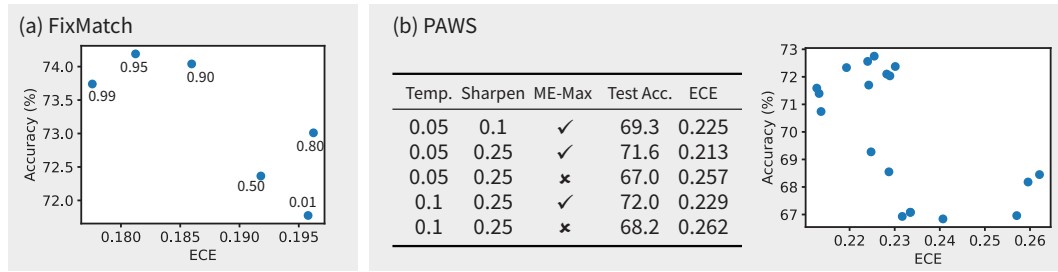

Figure 2: **Empirical study on CIFAR-100, 4000 labels.** (a) FixMatch: Test accuracy (%) against ECE (lower ECE is better calibration) when varying the threshold $\tau$ (value shown beside each scatter point). (b) PAWS: effect of various parameters on test accuracy and ECE; scatter plot explores a wider range of parameters, Temp. $\in [0.03, 0.1]$ and Sharpen $\in [0.1, 0.5]$.

where the latter is a regularization term $L_{\text{me-max}} = H(\bar{q})$ that seeks to maximize the entropy of the average of predictions $\bar{q} := (1/(2\mu B)) \sum_{u=1}^{\mu B} (\rho_t(q_{1,u}) + \rho_t(q_{2,u}))$.

**Calibration metrics.** A popular empirical metric to measure a model's calibration is via the *Expected Calibration Error (*ECE*)*. Following (Guo et al., 2017; Minderer et al., 2021), we focus on a slightly weaker condition and consider only the model's most likely class-prediction, which can be computed as follows. Let $\rho_0(q)$ denote the model's prediction (where $\rho_0$ is the argmax operation) as defined before, the model predictions on a batch of $N$ samples are grouped into $M$ equal-interval bins, i.e. $\mathcal{B}_m$ contains the set of samples with $\rho_0(q) \in (\frac{m-1}{M}, \frac{m}{M}]$. ECE is then computed as the expected difference between the accuracy and confidence of each bin over all $N$ samples:

$$\text{ECE} = \sum_{m=1}^{M} \frac{|\mathcal{B}_m|}{N} |\text{acc}(\mathcal{B}_m) - \text{conf}(\mathcal{B}_m)| \tag{3}$$

where $\text{acc}(\mathcal{B}_m) = (1/|\mathcal{B}_m|) \sum_{i \in \mathcal{B}_m} \mathbb{1}(\rho_0(q_i) = y_i)$ and $\text{conf}(\mathcal{B}_m) = (1/|\mathcal{B}_m|) \sum_{i \in \mathcal{B}_m} \max q_i$ with $y_i$ the true label of sample $i$. In this work, we estimate ECE using $M = 10$ bins. We also caveat here that while ECE is not free from biases (Minderer et al., 2021), we chose ECE over alternatives (Brier, 1950; DeGroot & Fienberg, 1983) due to its simplicity and widespread adoption.

## 4    CALIBRATION IN SEMI-SUPERVISED LEARNING

**Better calibration is correlated to better performances.** To motivate our work, we first perform detailed ablation studies on SOTA SSL methods. In particular, we used FixMatch (Sohn et al., 2020) and PAWS (Assran et al., 2021), each from the two families, due to their strong performance. FixMatch relies on using a high value of the selection threshold $\tau$ to mitigate confirmation bias by accepting only the most credible pseudo-labels; on the other hand, PAWS uses multiple techniques that seem to have an effect on shaping the output prediction distribution and implicitly controlling calibration — these include label smoothing (Müller et al., 2019), mean-entropy maximization (Joulin & Bach, 2012), sharpening (Berthelot et al., 2019b;a; Xie et al., 2019a) and temperature scaling (Guo et al., 2017). In Fig. 2, we ablated on all of these parameters and observe an overall trend, for both methods, that model performance is strongly correlated to better calibration (i.e. lower ECE). However, these trends are inevitably noisy since none of these parameters tunes for calibration in isolation but are instead optimized towards performance. Therefore, in this work, we aim to explore techniques that predominantly adjusts for model calibration in these methods to clearly demonstrate the direct effect of improving calibration towards model accuracy.

### 4.1    IMPROVING CALIBRATION WITH BAYESIAN MODEL AVERAGING

Bayesian techniques have been widely known to produce well-calibrated uncertainty estimates (Wilson & Izmailov, 2020), thus in our work, we explored the use of approximate Bayesian Neural Networks (BNN). To minimize the additional computational overhead, we propose to only replace the **final layer** of the network with a BNN layer. We propose a novel family of SSL methods termed BAM- (for BAyesian Model averaging) and provide an illustration of this approach in Fig. 1.

For threshold-mediated methods the BNN layer is the linear classification head $h$ and for representation learning methods this is the final layer of the projection head. For brevity we will simply denote this BNN layer to be $h$ and an input embedding to this layer to be $v$ in this section. Following a Bayesian approach, we first assume a prior distribution on weights $p(\theta_h)$. Given some training data $\mathcal{D}_\mathcal{X} := (X, Y)$, we seek to calculate the posterior distribution of weights, $p(\theta_h|\mathcal{D}_\mathcal{X})$, which can then be used to derive the posterior predictive $p(y|v, \mathcal{D}_\mathcal{X}) = \int p(y|v, \theta_h)p(\theta_h|\mathcal{D}_\mathcal{X})d\theta_h$. This process is also known as "Bayesian model averaging" or "Bayesian marginalization" (Wilson & Izmailov, 2020). Since exact Bayesian inference is computationally intractable for neural networks, we adopt a variational approach following (Blundell et al., 2015), where we learn a Gaussian variational approximation to the posterior $q_\phi(\theta_h|\phi)$, parameterized by $\phi$, by maximizing the evidence lower-bound (ELBO) (see Appendix C.1 for details). The ELBO $= \mathbb{E}_q \log p(Y|X; \theta) - KL(q(\theta|\phi)\|p(\theta))$ consists of a log-likelihood (data-dependent) term and a KL (prior-dependent) term.

**Theoretical results.** We motivate our approach by using Corollary 1 below, derived from the PAC-Bayes framework (see Appendix B for the proof). The statement shows the generalization error bounds on the variational posterior in the SSL setting and suggests that this generalization error is upper bounded by the negative ELBO. This motivates our approach, i.e. by maximizing the ELBO we improve generalization by minimizing the upper bound to the generalization error. The second term on the right side of the inequality characterises the SSL setting, and vanishes in the supervised setting.

**Corollary 1** *Let $\mathcal{D}$ be a data distribution where i.i.d. training samples are sampled, of which $N_l$ are labeled, $(x, y) \sim \mathcal{D}^{N_l}$ and $N_u$ are unlabeled, $(x, \hat{y}) \sim \mathcal{D}^{N_u}$ where $\hat{y}_i$ ($y_i$) denotes the model-assigned pseudo-labels (true labels) for input $x_i$. For the negative log likelihood loss function $\ell$, assuming that $\ell$ is sub-Gaussian (see Germain et al. (2016)) with variance factor $s^2$, then with probability at least $1 - \delta$, the generalization error (denoted as $\mathcal{L}_\mathcal{D}^\ell$) of the variational posterior $q(\theta|\phi)$ is given by,*

$$\mathbb{E}_{q(\theta|\phi)}\mathcal{L}_\mathcal{D}^\ell(q) \leq \frac{1}{N}[-\text{ELBO}] - \mathbb{E}_{q(\theta|\phi)}\left[\frac{1}{N_u}\sum_{i=1}^{N_u}\log\frac{p(\hat{y}_i|x_i; \theta)}{p(y_i|x_i; \theta)}\right] + \frac{1}{N}\log\frac{1}{\delta} + \frac{s^2}{2} \quad (4)$$

As depicted in Fig. 1, pseudo-labeling proceeds in two-stages: 1) $M$ weights from the BNN layer are sampled and predictions are derived from the Monte Carlo estimated posterior predictive, i.e. $\hat{q} = (1/M)\sum_m^M h(v, \theta_h^{(m)})$, and 2) the selection criteria, if present, is based upon their variance, $\sigma_c^2 = (1/M)\sum_m^M(h(v, \theta_h^{(m)}) - \hat{q})^2$, at the predicted class $c = \text{argmax}_{c'}[\hat{q}]_{c'}$. This constitutes a better uncertainty measure as compared to the maximum logit value used in threshold-mediated methods and is highly intuitive — if the model's prediction has a large variance, it is highly uncertain and the pseudo-label should not be accepted. In practice, as $\sigma_c^2$ decreases across training, we use a simple quantile $Q$ over the batch to define the threshold where pseudo-labels of samples with $\sigma_c^2 < Q$ are accepted, with $Q$ as a hyperparameter (see Appendix C.1 for pseudocode). In representation learning methods where predictions are computed using a non-parametric nearest neighbour classifier, we use the "Bayesian marginalized" versions of the representations, i.e. $q_i = \pi_d(\hat{z}_i, \{\hat{z}_s\})$ in Eq. (2) where $\hat{z} = (1/M)\sum_m^M h(v, \theta_h^{(m)})$. We explore the effectiveness of BAM by modifying upon SOTA SSL methods and name them "BAM-X", which incorporates approximate BAyesian Model averaging (BAM) during pseudo-labeling for SSL method X.

## 4.2 IMPROVING CALIBRATION WITH WEIGHT AVERAGING TECHNIQUES

A BNN classifier has two desirable characteristics: 1) multiple predictions are "Bayesian marginalized" and 2) better measure of uncertainty (i.e. confidence estimates) using the variance across predictions. Representation learning SSL methods that do not use a selection metric cannot explicitly benefit from (2) and in these cases, instead of aggregating over just one layer, one could seek to aggregate over the entire network. To explore the generality of the importance of calibration in SSL beyond a Bayesian layer approach, we further explored well-established weight averaging approaches such as Stochastic Weight Averaging (Izmailov et al., 2018) (SWA) and Exponential Moving Averaging (EMA) (Tarvainen & Valpola, 2017; He et al., 2020; Grill et al., 2020a) and studied their role in calibration during pseudo-labeling.

To do so, we maintain a separate set of non-trainable weights $\theta_g$ containing the aggregated weight average which are used throughout training to produce better calibrated pseudo-labels. In SWA,

these weights are updated via $\theta_g \leftarrow (n_a\theta_g + \theta_f)/(n_a + 1)$ at every iteration, where $n_a$ represents the total number of models in the aggregate and $\theta_f$ are the trainable parameters of our backbone encoder as before. The update of EMA only differs slightly: $\theta_g \leftarrow \gamma\theta_g + (1-\gamma)\theta_f$, where $\gamma$ is a momentum hyperparameter controlling how much memory $\theta_g$ should retain at each iteration (see Appendix C.2 for pseudocode). While EMA has been previously explored in the context of SSL (Tarvainen & Valpola, 2017), to the best of our knowledge SWA has not been explored in SSL and more importantly, their link to pseudo-label calibration has not been explicitly shown. In this work, we demonstrate that the effectiveness of such weight averaging approaches can be understood from an improved calibration of pseudo-labels.

## 5 EXPERIMENTAL SETUP

In all our experiments, we begin with and modify upon the original implementations of the baseline SSL methods. The backbone encoder $f$ is a Wide ResNet-28-2, Wide ResNet-28-8 and ResNet-50 for the CIFAR-10, CIFAR-100 and ImageNet benchmarks respectively. Where possible, the default hyperparameters and dataset-specific settings (learning rates, batch size, optimizers and schedulers) recommended by the original authors were used and where unavailable (for e.g. CIFAR-100 on PAWS), the baseline was first optimized to the best we could. To ensure fairness, we use the same configuration and same set of hyperparameters when comparing our methods against the baselines.

**Bayesian model averaging with a BNN final layer.** We set the weight priors as unit Gaussians and use a separate Adam optimizer for the BNN layer with learning rate 0.01, no weight decay and impose the same cosine learning rate scheduler as the backbone. We set $Q = 0.75$ for the CIFAR-100 benchmark and $Q = 0.95$ for the CIFAR-10 benchmark; which are both linearly warmed-up from 0.1 in the first 10 epochs. As $Q$ is computed across batches, we improve stability by using a moving average of the last 50 thresholds. For representation learning methods without a parametric classifier, we do not use a separate optimizer and simply impose a one-minus-cosine scheduler (i.e. scheduler (2) from the next paragraph) for the coefficient to the KL-divergence loss that goes from 0 to 1 in $T$ epochs where we simply picked the best from $T \in \{50, 100\}$.

**Weight averaging.** We delay the onset of SWA to after some amount of training has elapsed (i.e. $T_{swa}$ epochs) since it is undesirable to include the randomly initialized weights to the aggregate. We set $T_{swa} = 200$ and $T_{swa} = 100$ for CIFAR-10 and CIFAR-100 respectively. In contrast, EMA has a natural curriculum to "forget" older model parameters since more recent parameters are given more weight in the aggregate. We experimented with two schedules for $\gamma$: 1) using a linear warm-up from 0 to 0.996 in 50 epochs and then maintaining $\gamma$ at 0.996 for the rest of training and 2) using a one-minus-cosine scheduler starting from 0.05 and decreasing to 0 resulting in a 0.95 to 1 range for $\gamma$. We visualize these schedulers and include ablation studies on them and on $T_{\text{swa}}$ in Appendix G.3.

**ECE and test accuracy evaluation.** In our experiments, we found that the test accuracy exhibits a considerable amount of noise across training, especially in label-scarce settings. Sohn et al. (2020) proposes to take the median accuracy of the last 20 checkpoints, while Zhang et al. (2021) argues that this fixed training budget approach is not suitable when convergence speeds of the algorithms are different (as the faster converging algorithm would over-fit more severely at the end) and thus report also the overall best accuracy. In our experiments, we adopt a balance between the two aforementioned approaches: we consider the median of 20 checkpoints around the best accuracy checkpoint as the *convergence criteria*, and report this value as the test accuracy. The ECE is reported when the model reaches this convergence criteria (One could also also aggregate the ECEs up till convergence — we found this gave similar trends and thus report the simpler metric.) For PAWS, we follow Assran et al. (2021) and report the test accuracy and ECE from the soft nearest neighbour classifier on the CIFAR benchmarks and after fine-tuning a linear head for the ImageNet benchmark.

## 6 RESULTS

**Bayesian model averaging.** Our main results on CIFAR-10 and CIFAR-100 in Table 1 show the use of a BNN layer to improve calibration for all methods in our study. Table 1 demonstrates that incorporating a BNN final layer successfully reduces the ECE over the baselines across all the

Table 1: **Calibration in SSL** showing "Test accuracy (%) / ECE". Improving calibration improves test accuracies consistently across all benchmarks and across a variety of SSL methods, Pseudolabel (PL) (Lee, 2013), FixMatch (FM) (Sohn et al., 2020), UDA (Xie et al., 2019a) and PAWS (Assran et al., 2021). Rows with the prefix "BAM-" refers to BAyesian model averaging via a BNN final layer, while +SWA and +EMA are using weight averaging techniques. For each benchmark, results are averaged over 3 random dataset splits. Entries with dashes (-) indicate that training did not converge.

| | CIFAR-10 | | CIFAR-100 | | |
|---|---|---|---|---|---|
| | 250 labels | 2500 labels | 400 labels | 4000 labels | 10000 labels |
| **Threshold-mediated methods** | | | | | |
| PL (repro) | 53.8 / 0.395 | 81.0 / 0.197 | - | 49.1 / 0.360 | 66.4 / 0.231 |
| BAM-PL (ours) | - | 81.5 (↑0.5) / 0.190 | - | 50.8 (↑1.7) / 0.342 | 66.4 / 0.229 |
| FM (repro) | 94.2 / 0.051 | 95.7$_{\pm0.03}$ / 0.039$_{\pm0.0}$ | 56.4$_{\pm1.6}$ / 0.366$_{\pm0.017}$ | 74.2$_{\pm0.2}$ / 0.183$_{\pm0.003}$ | 78.1$_{\pm0.2}$ / 0.147$_{\pm0.001}$ |
| BAM-FM (ours) | 95.0 (↑0.8) / 0.045 | 95.7$_{\pm0.1}$ / 0.039$_{\pm0.0}$ | 59.0$_{\pm1.4}$ (↑2.6) / 0.331$_{\pm0.015}$ | 74.8$_{\pm0.09}$ (↑0.6) / 0.171$_{\pm0.002}$ | 78.1$_{\pm0.2}$ / 0.139$_{\pm0.002}$ |
| UDA (repro) | 94.3 / 0.050 | 95.7 / 0.039 | 44.1$_{\pm0.7}$ / 0.473$_{\pm0.013}$ | 72.9$_{\pm0.01}$ / 0.189$_{\pm0.003}$ | 77.2$_{\pm0.3}$ / 0.154$_{\pm0.002}$ |
| BAM-UDA (ours) | **95.3** (↑1.0) / 0.042 | **95.8** (↑0.1) / 0.038 | **60.3**$_{\pm0.6}$ (↑16.2) / 0.314$_{\pm0.005}$ | **75.2**$_{\pm0.1}$ (↑2.3) / 0.165$_{\pm0.002}$ | 78.3$_{\pm0.2}$ (↑1.1) / 0.138$_{\pm0.003}$ |
| **Rep-learning methods** | | | | | |
| PAWS (repro) | 90.5 / 0.091 | 95.3 / 0.046 | 44.5 / 0.455 | 71.5$_{\pm0.3}$ / 0.232$_{\pm0.005}$ | 75.6$_{\pm0.1}$ / 0.198$_{\pm0.002}$ |
| BAM-PAWS (ours) | 94.5 (↑4.0) / 0.054 | 95.4 (↑0.1) / 0.046 | 47.4 (↑2.9) / 0.372 | 72.5 (↑1.0) / 0.228 | 76.5 (↑0.9) / 0.195 |
| PAWS+SWA (ours) | 90.9 (↑0.4) / 0.088 | 95.6 (↑0.3) / 0.046 | 45.2 (↑0.7) / 0.444 | 74.5$_{\pm0.3}$ (↑3.0) / 0.193$_{\pm0.002}$ | **78.4**$_{\pm0.4}$ (↑2.8) / 0.164$_{\pm0.003}$ |
| PAWS+EMA (ours) | 90.9 (↑0.4) / 0.087 | **95.8** (↑0.5) / 0.042 | 46.2 (↑1.7) / 0.445 | 73.5$_{\pm0.2}$ (↑2.0) / 0.193$_{\pm0.005}$ | 77.2$_{\pm0.1}$ (↑1.6) / 0.168$_{\pm0.003}$ |

benchmarks and as a result, we also attained significant improvements in test accuracies, notably up to 16.2% (for UDA on CIFAR-100-400 labels). Interestingly, while the baseline FixMatch outperforms UDA across all the benchmarks, improving calibration in UDA (i.e. BAM-UDA) allows it to outperfom FixMatch and even the calibrated version of FixMatch (i.e. BAM-FM). A key difference between FixMatch and UDA is the use of hard pseudo-labels in FixMatch (i.e. $t \rightarrow 0$ in $\rho_t$ defined in Section 3) versus soft pseudo-labels in UDA (with $t = 0.4$); this suggests that a Bayesian classifier is more effective in conjunction with soft pseudo-labels. Intuitively, this makes sense, since better calibration could potentially allow the model to leverage on information about its prediction on classes apart from the one it's most confident of. This further underscores the importance of calibration — while information about the model's prediction on other classes is useful, the lack of proper calibration prohibits the model from using this information, resulting in stronger confirmation bias compared to the case where this information is not used (i.e. UDA vs FixMatch).

Leveraging on this insight, we significantly reduce the sharpening of the pseudo-labels in BAM-UDA by setting $t = 0.9$ (see Appendix G.2 for ablations on $t$). BAM-UDA *consistently outperforms* all threshold-mediated baselines across all CIFAR benchmarks . Overall, we found that the improvements in both calibration and test accuracy are more significant for label-scarce settings, as expected, since the problem of confirmation bias is more acute there and thus calibration can provide greater benefits by mitigating this. The additional computation overhead incurred from our methods is minimal, adding approximately only 2-5% in wall-clock time (see Appendix H). While previous works (Sohn et al., 2020) also include extreme low label settings such as CIFAR-10-40 labels, we found this benchmark to be highly sensitive to random initialization and thus exclude them in our study.

**Why does improving calibration improve performance?** To further understand the dynamics between calibration and test accuracies, we track the test accuracy and ECE over the course of training for UDA, FixMatch and BAM-UDA and plot them in Fig. 3. While the baselines learn effectively for the initial stages of training, learning is eventually hindered — as the model is encouraged to output increasingly confident predictions due to the entropy minimization objective, in the absence of explicit calibration, the baselines inevitably become over-confident, resulting in a drop in test accuracy. This phenomenon of confirmation bias is evident from that 1) the baselines' output predictions become increasingly over-confident (Fig. 3d) and 2) they make increasingly more mistakes on the accepted pseudo-labels (Fig. 3c). In contrast, explicit calibration via Bayesian model averaging allows BAM-UDA to improve its calibration throughout training, resulting in a constantly improving purity rate and effectively mitigating confirmation bias, thus promoting learning for longer periods to result in better final performances. Further ablation studies are discussed in Appendix G.2.

**Weight averaging techniques.** The effect of better calibration is also clearly demonstrated in the weight averaging approaches (Fig. 4). This is particularly evident through PAWS-SWA—when SWA was switched on (at $T_{\mathrm{swa}} = 200$ epochs), the ECE quickly dips and the test accuracy also correspondingly spikes. Furthermore, we also observe a significant improvement in convergence rates resulting from the weight averaging techniques, i.e. better test accuracies can be attained in just a third of the number of training iterations of the baseline. Ablations on $T_{\mathrm{swa}}$ and momentum

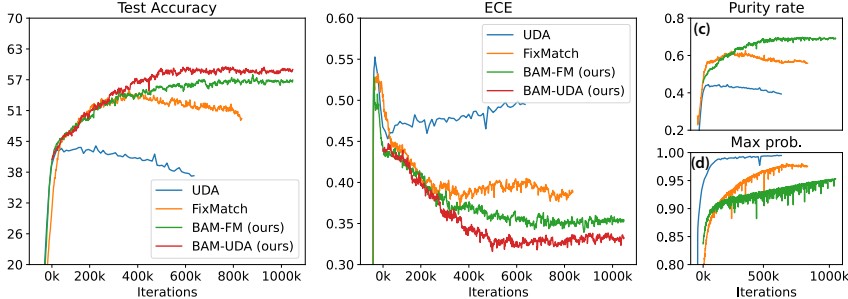

Figure 3: **UDA, FixMatch, BAM-FM and BAM-UDA across training** on CIFAR-100 with 400 labels (a) Test accuracies and (b) ECE as a function of training time. (c) Purity rate shows the accuracy rate of unlabeled samples that are accepted by the selection metric and (d) Max prob shows the model's confidence, i.e. the average maximum class probability of all unlabeled samples.

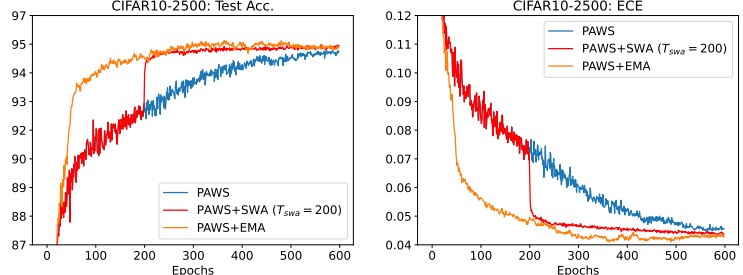

Figure 4: **Weight averaging techniques on PAWS for CIFAR-10 with 2500 labels** displays highly consistent relationships between increasing test accuracy and decreasing ECE.

schedules are presented in Appendix G.3. Our motivation for exploring weight averaging techniques in PAWS was attributed to its lack of a selection metric; however, given that almost all SSL methods have a "stop-gradient" operation on pseudo-labels, one could potentially also explore the use of weight averaging techniques in threshold-mediated methods. Results are shown in Appendix G.1, where we studied the use of EMA on the best performing threshold-mediated baseline, i.e. FixMatch. Notably, we found that EMA consistently under-performs Bayesian model averaging, highlighting the superiority of a BNN classifier in threshold-mediated methods. This could be attributed to the fact that, in addition to averaging predictions, a BNN classifier provides better uncertainty estimates to improve the selection metric in these family of methods (see Appendix G.1 for further discussion).

## 6.1 LARGE-SCALE & REAL-WORLD DATASETS

**ImageNet.** We further verified the effectiveness of calibration on the large-scale dataset ImageNet. Out of all the baseline methods in our study, PAWS reports the best performance on the ImageNet-10% benchmark (i.e. 73.9% for 100 epochs vs 71.5% on the best performing threshold-mediated method, FixMatch). Due to computational resource limitations, we focus our experiments only on the best baseline PAWS and explored calibration improvements using SWA and EMA. We use the 10% labels setting of Assran et al. (2021) with slight modifications to the default setting in order to fit on our system (see Appendix D.2 for implementation details). In particular, we maintain the exact same configuration across the baseline and our methods. Results are shown in Table 2, where our methods provide consistent improvements in calibration, and correspondingly, we also obtain consistent improvements on the test accuracy.

Table 2: **ImageNet-10%** showing "Top-1 test accuracy (%) / ECE" for PAWS and our methods, trained for 200 epochs and fine-tuned with a linear head.

|  | IN-10% |
| --- | --- |
| PAWS | 74.4 / 0.186 |
| +SWA (ours) | 74.5 / 0.186 |
| +EMA (ours) | **75.1** / 0.183 |

**Long-tailed datasets.** Datasets in the real world are often long-tailed or class-imbalanced, where some classes are more commonly observed while others are rare. We curate long-tailed versions from the CIFAR datasets following Cao et al. (2019), where $\alpha$ indicates the imbalance ratio (i.e. ratio between the sample sizes of the most frequent and least frequent classes). We randomly select 10%

Table 3: **Long-tailed CIFAR-10 & CIFAR-100** showing "Test accuracy (%) / ECE". We use 10% of labels from each class. For better interpretation, we also show the supervised (100% labels) accuracy reported in Cao et al. (2019), which use a different architecture, ResNet-32, and an algorithm targeted for long-tailed problems.

| | CIFAR-10-LT | | CIFAR-100-LT | |
|---|---|---|---|---|
| | $\alpha = 10$ | $\alpha = 100$ | $\alpha = 10$ | $\alpha = 100$ |
| FM | 91.3 / 0.073 | 70.0 / 0.26 | 48.8 / 0.38 | 28.6 / 0.55 |
| BAM-UDA (ours) | **91.6** (↑0.3) / 0.067 | **71.2** (↑1.2) / 0.24 | **53.6** (↑4.8) / 0.32 | **31.9** (↑3.3) / 0.50 |
| Supervised | 88.2 | 77.0 | 58.7 | 42.0 |

of the samples in each class to form the labeled set; see further details in Appendix E. We use the best performing baseline method from Table 1, i.e. FixMatch (FM), as our baseline and compare against BAM-UDA. Results are shown in Table 3, where BAM-UDA achieves consistent improvements over FM in both calibration and accuracy across all benchmarks. Notably, gains from improving calibration are more significant than those in the class-balanced settings (for e.g. BAM-UDA improves upon FM by a smaller margin of 1.5% on the CIFAR-100-4000 labels benchmark which is also approximately 10%), further underscoring the utility of improving calibration in more challenging SSL problems. Further results are in Appendix E.2, where test samples were separated into three groups depending on the number of samples per class and test accuracies are plotted for each group.

**Photonics science.** A practical example of a real-world domain where long-tailed datasets are prevalent is that of science – samples with the desired properties are often much rarer than trivial samples. Further, SSL is highly important in scientific domains since labeled data is particularly scarce (owing to the high resource cost needed for data collection). To demonstrate the effectiveness of calibration, we adopt a problem in photonics (Loh et al., 2022), where the task is

Table 4: **Photonic crystals (PhC) band gap prediction** showing "Test accuracy (%) / ECE". The fully-supervised (100% labels) accuracy is 88.5%.

| | PhC-10% | PhC-1% |
|---|---|---|
| FM | 78.8 / 0.098 | 55.0 / 0.385 |
| BAM-UDA | **81.0** (↑2.2) / 0.052 | **56.9** (↑1.9) / 0.356 |

a 5-way classification of photonic crystals (PhCs) based on their band gap sizes. A brief summary and visualization of this dataset are detailed in Appendix F. We explored an approach similar to FixMatch for the baseline and similar to BAM-UDA for ours (with modifications needed in the augmentation strategies; see Appendix F). Results are shown in Table 4, where we demonstrate the consistency of BAM-UDA's effectiveness in improving calibration and accuracies in this real-world problem.

# 7 CONCLUSION

In this work, we demonstrate that calibration plays a crucial role in SSL methods – specifically, since confirmation bias is a fundamental problem in SSL, it is imperative for the model to be well-calibrated to mitigate this problem. In particular, we demonstrated the strong correspondence between calibration and model performance in SSL methods and proposed to use approximate Bayesian techniques to directly improve calibration in state-of-the-art SSL methods. Notably, we demonstrate their effectiveness across a broad range of SSL methods and further underscore their importance in more challenging real-world datasets. We hope that our findings can motivate future research directions to incorporate techniques targeted for optimizing calibration during the development of new SSL methods. Furthermore, while the primary goal of improving calibration is to mitigate confirmation bias during pseudo-labeling, an auxiliary benefit brought about by our approach is a better calibrated network, i.e. one that can better quantify its uncertainty, which is highly important for real-world applications. A potential limitation in our work lies in the use of ECE as a metric to measure calibration which, while commonly used across literature, are not free from flaws (Nixon et al., 2019). However, in our work, we empirically demonstrate that despite their flaws, the ECE metric still provides good correlations to measuring confirmation bias and test accuracy.

## 8 REPRODUCIBILITY STATEMENT

To ensure that our reported results are reproducible, all source code will be made available in a publicly hosted repository soon. We further list all details on training configurations and hyperparameter settings in Section 5 and Appendix D.

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

# A   COMPARISON WITH PRIOR ART IN SSL

Table 5: **Comparison of techniques used in BAM- with techniques used in prior art of SSL.**

| SSL method | Augmentation | Pseudo-label | Selection metric |
|---|---|---|---|
| Temporal ensemble (Laine & Aila, 2016) | Weak | model from earlier step | - |
| Mean teacher (Tarvainen & Valpola, 2017) | Weak | EMA | - |
| UDA (Xie et al., 2019a) | Weak & strong | Sharpen | Logit thresholding |
| MixMatch (Berthelot et al., 2019b) | Weak | Averaging aug + sharpen | - |
| FixMatch (Sohn et al., 2020) | Weak & strong | Hard labels | Logit thresholding |
| FlexMatch (Zhang et al., 2021) | Weak & strong | Hard labels | Class-wise logit threshold |
| Our main method BAM- | Weak & strong | **Posterior sampling** + sharpen | **Variance thresholding** |

# B   PROOF OF COROLLARY 1

PAC-Bayesian theory aims to provide bounds on the generalization risk under the assumption that samples are i.i.d.. While PAC-Bayesian bounds typically apply to bounded losses, Germain et al. (2016) extends them to unbounded losses (which is necessary in our work since it uses the unbounded log-loss). Their theoretical result is reproduced below in Theorem 1, under assumptions for sub-Gaussian losses. These bounds commonly assume the supervised setting; in this proof, our goal is to extend them to the semi-supervised learning setting where in addition to labels from an empirical set, our loss is also mediated by pseudo-labels assigned by the model.

Let $(X, Y) \sim \mathcal{D}^N$ denote $N$ i.i.d. training samples from the data distribution $\mathcal{D}$, where $(X, Y) = \left\{ \{(x_i, y_i)\}_{i=1}^{N_l}, \{(x_i, \hat{y}_i)\}_{i=1}^{N_u} \right\}$ represents the set of $N_l$ labeled input-output pairs and $N_u$ pairs of unlabeled input and their model-assigned pseudo-labels $\hat{y}$. Here, $N_l + N_u = N$. Let the loss function be $\ell(f, x, y) \to \mathbb{R}$, the generalization risk on distribution $\mathcal{D}$, i.e. $\mathcal{L}_{\mathcal{D}}^{\ell}(f)$, and the empirical risk on the training set, i.e. $\mathcal{L}_{X,Y}^{\ell}(f)$, is given by:

$$\mathcal{L}_{\mathcal{D}}^{\ell}(f) = \mathop{\mathbb{E}}_{(x,y) \sim \mathcal{D}} \ell(f, x, y); \quad \mathcal{L}_{X,Y}^{\ell}(f) = \frac{1}{N_l} \sum_{i=1}^{N_l} \ell(f, x_i, y_i) + \frac{1}{N_u} \sum_{i=1}^{N_u} \ell(f, x_i, \hat{y}_i)$$

On assumptions that the loss is sub-Gaussian (see Boucheron et al. (2013) section 2.3), i.e. a loss function $\ell$ is sub-Gaussian with variance $s^2$ under a prior $\pi$ and $\mathcal{D}$ if it can be described by a sub-Gaussian random variable $v = \mathcal{L}_{\mathcal{D}}^{\ell}(f) - \ell(f, x, y)$, the generalization error bounds are given by Theorem 1 (Germain et al., 2016) below.

**Theorem 1** *(Corollary 4 from Germain et al. (2016)) Let $\pi$ be the prior distribution and $\hat{\rho}$ be the posterior. If the loss is sub-Gaussian with variance factor $s^2$, with probability at least $1 - \delta$ over the choice of $(X, Y) \sim \mathcal{D}^N$,*

$$\mathop{\mathbb{E}}_{f \sim \hat{\rho}} \mathcal{L}_{\mathcal{D}}^{\ell}(f) \leq \mathop{\mathbb{E}}_{f \sim \hat{\rho}} \mathcal{L}_{X,Y}^{\ell}(f) + \frac{1}{N} \left( KL(\hat{\rho} \| \pi) + \log(1/\delta) \right) + \frac{1}{2} s^2$$

Beginning from Theorem 1, the generalization error of the variational posterior $q(\theta|\phi)$ under the negative log likelihood loss and prior $p(\theta)$ in our SSL setting can be found by replacing $\hat{\rho}$ with $q(\theta|\phi)$

and $\ell$ with $\ell^{nll}(f, x_i, y_i) := -\log p(y_i|x_i; \theta)$;

$$\mathbb{E}_{f \sim q} \mathcal{L}_{\mathcal{D}}^{\ell}(f) \leq \mathbb{E}_{f \sim q} \mathcal{L}_{X,Y}^{\ell}(f) + \frac{1}{N}\left(KL(q(\theta|\phi)\|p(\theta)) + \log(1/\delta)\right) + \frac{1}{2}s^2$$

$$= \mathbb{E}_q\left[\frac{1}{N_u}\sum_{i=1}^{N_u} -\log p(\hat{y}_i|x_i; \theta) + \frac{1}{N_l}\sum_{i=1}^{N_l} -\log p(y_i|x_i; \theta)\right]$$
$$+ \frac{1}{N}\left(KL(q(\theta|\phi)\|p(\theta)) + \log(1/\delta)\right) + \frac{1}{2}s^2$$

$$= \mathbb{E}_q\left[\frac{1}{N_u}\sum_{i=1}^{N_u} -\log\frac{p(\hat{y}_i|x_i; \theta)}{p(y_i|x_i; \theta)} + \frac{1}{N_u}\sum_{i=1}^{N_u} -\log p(y_i|x_i; \theta) + \frac{1}{N_l}\sum_{i=1}^{N_l} -\log p(y_i|x_i; \theta)\right]$$
$$+ \frac{1}{N}\left(KL(q(\theta|\phi)\|p(\theta)) + \log(1/\delta)\right) + \frac{1}{2}s^2$$

$$= \mathbb{E}_q\left[\frac{1}{N_u}\sum_{i=1}^{N_u} -\log\frac{p(\hat{y}_i|x_i; \theta)}{p(y_i|x_i; \theta)} + \frac{1}{N}\sum_{i=1}^{N} -\log p(y_i|x_i; \theta)\right] + \frac{1}{N}\left(KL(q(\theta|\phi)\|p(\theta)) + \log(1/\delta)\right) + \frac{1}{2}s^2$$

$$= \mathbb{E}_q\left[-\frac{1}{N_u}\sum_{i=1}^{N_u}\log\frac{p(\hat{y}_i|x_i; \theta)}{p(y_i|x_i; \theta)}\right] + \mathbb{E}_q\left[-\frac{1}{N}\log p(Y|X; \theta)\right] + \frac{1}{N}\left(KL(q(\theta|\phi)\|p(\theta)) + \log(1/\delta)\right) + \frac{1}{2}s^2$$

$$\mathbb{E}_{f \sim q} \mathcal{L}_{\mathcal{D}}^{\ell}(f) \leq \mathbb{E}_q\left[-\frac{1}{N_u}\sum_{i=1}^{N_u}\log\frac{p(\hat{y}_i|x_i; \theta)}{p(y_i|x_i; \theta)}\right] + \frac{1}{N}\left[-\text{ELBO}\right] + \frac{1}{N}\log(1/\delta) + \frac{1}{2}s^2$$

where we define $\text{ELBO} = \mathbb{E}_{q(\theta|\phi)}\left[\log p(Y|X; \theta)\right] - KL(q(\theta \mid \phi)\|P(\theta))$. In the next section, Appendix C.1, we will see that this is the term we are maximizing in our loss objective. By maximizing the ELBO, i.e. minimizing the negative ELBO, we are minimizing the upper bound to the generalization error of our variational posterior. The first term in the last line adds a divergence measure between the pseudo-label prediction distribution and the ground truth distribution — in the fully supervised setting $\hat{y}_i \to y_i$ and this term vanishes.

## C   FORMULATION DETAILS AND PSEUDOCODE OF OUR METHODS

### C.1   BAYESIAN MODEL AVERAGING VIA A BNN FINAL LAYER

Following the notations in Section 4.1, we denote the BNN layer to be $h$ and an input embedding to this layer to be $v$ in this section. We assume a prior distribution on weights $P(\theta_h)$ and seek to calculate the posterior distribution of weights given the empirical/training data, $P(\theta_h|\mathcal{D}_{\mathcal{X}})$, where $\mathcal{D}_{\mathcal{X}} := (X, Y)$, which can then be used to compute the posterior predictive during inference. As exact Bayesian inference is intractable for neural networks, we adopt a variational approach following (Blundell et al., 2015) to approximate the posterior with a Gaussian distribution parameterized by $\phi$, $q(\theta|\phi)$. From now, we will drop the $h$ index in $\theta_h$ for brevity. To learn the variational approximation, we seek to minimize the Kullback-Leibler (KL) divergence between the Gaussian variational approximation and the posterior:

$$\phi^* = \text{argmin}_\phi KL\left(q(\theta|\phi) \,\|\, p(\theta|X, Y)\right)$$
$$= \text{argmin}_\phi \int q(\theta|\phi)\log\frac{q(\theta|\phi)}{p(\theta|X, Y)}d\theta$$
$$= \text{argmin}_\phi \mathbb{E}_{q(\theta|\phi)}\log\frac{q(\theta|\phi)p(Y|X)}{p(Y|X; \theta)p(\theta)}$$
$$= \text{argmin}_\phi \mathbb{E}_{q(\theta|\phi)}\left[\log q(\theta|\phi) - \log p(Y|X; \theta) - \log p(\theta)\right] + \log p(Y|X)$$
$$= \text{argmin}_\phi KL(q(\theta|\phi)\|p(\theta)) - \mathbb{E}_{q(\theta|\phi)}\log p(Y|X; \theta) + \log p(Y|X)$$
$$= \text{argmin}_\phi \left([-\text{ELBO}] + \log p(Y|X)\right)$$

where in the last line, $\text{ELBO} = \mathbb{E}_{q(\theta|\phi)}\left[\log p(Y|X; \theta)\right] - KL(q(\theta|\phi)\|p(\theta))$ is the evidence lower-bound which consists of a log-likelihood (data-dependent) term and a KL (prior-dependent) term.

---

**Algorithm 1** PyTorch-style pseudocode for Bayesian model averaging in UDA or FixMatch.

```
# f: backbone encoder network
# h: bayesian classifier
# KL_loss: KL term in evidence lower-bound
# H: cross-entropy loss
# Q: quantile parameter
# num_samples: number of weight samples
# method: `UDA` or `FM`
# shp: sharpen operation

threshold_list = []
for (xl, labels), xu in zip(labeled_loader, unlabeled_loader):
    x_lab, x_uw, x_us = weak_augment(xl), weak_augment(xu), strong_augment(xu)
    z_lab, z_uw, z_us = f(x_lab), f(x_uw), f(x_us) # get representations
    mean_uw, std_uw = bayes_predict(h, z_uw)
    threshold_list.pop(0) if len(threshold_list) > 50 # keep 50 most recent thresholds
    threshold_list.append(quantile(std_uw, Q))
    mask = std_uw.le(threshold_list.mean())

    # compute losses
    loss_kl = KL_loss(h) # prior-dependent (data-independent) loss
    loss_lab = H(h(z_lab), labels)
    if method == `UDA`:
        loss_unlab = H(h(z_us), shp(mean_uw)) * mask # sharpened soft pseudo-labels
    elif method == `FM`:
        loss_unlab = H(h(z_us), mean_uw.argmax(-1)) * mask # hard pseudo-labels
    loss = loss_lab + loss_unlab + loss_kl
    loss.backward()
    optimizer.step()

def bayes_predict(h, z):
    outputs = stack([h(z).softmax(-1) for _ in range(num_samples)]) # sample weights
    return outputs.mean(), outputs.std() # mean and std of predictions
```

---

Since $\mathrm{KL}\left(q(\theta|\phi) \,\|\, p(\theta|\mathcal{D}_\mathcal{X})\right)$ is intractable, we maximize the ELBO which is equivalent to minimizing the former up to the constant, $\log p(Y|X)$.

Each variational posterior parameter of the Gaussian distribution, $\phi$, consists of the mean ($\mu$) and the standard deviation (which is parametrized as $\sigma = \log(1 + \exp(\rho))$ so that $\sigma$ is always positive (Blundell et al., 2015)), i.e. $\phi = (\mu, \rho)$. To obtain a sample of the weights $\theta$, we use the reparametrization trick (Kingma et al., 2015) and sample $\epsilon \sim \mathcal{N}(0, I)$ to get $\theta = \mu + \log(1 + \exp(\rho)) \circ \epsilon$ where $\circ$ denotes elementwise multiplication. In other words, we double the number of learnable parameters in the layer compared to a non-Bayesian approach; however, this does not add a huge computational cost since only the last layer is Bayesian and the dense backbone remains non-Bayesian.

Algorithm 1 shows the PyTorch-style pseudo-code for BAM- in threshold-mediated methods (here showing asymmetric augmentation applicable for UDA or FixMatch). The main modifications upon the baseline methods include 1) computation of an additional KL loss term (between two Gaussians, i.e. the variational approximation and the prior), 2) taking multiple samples of weights to derive predictions and 3) replacing the acceptance criteria from using maximum probability to using standard deviation of predictions.

## C.2 Algorithm for SWA & EMA in PAWS

Algorithm 2 shows the pseudocode for the implementation of PAWS+SWA and PAWS+EMA in PyTorch. For brevity, we leave out details of the multicrop strategy, mean entropy maximization regularization and soft nearest neighbour classifier formulation which are all replicated from the original implementation. We defer readers to the original paper (Assran et al., 2021) for these details.

---

**Algorithm 2** PyTorch-style pseudocode for PAWS-SWA and PAWS-EMA.

---

```
# f: backbone encoder network
# g: weight aggregated encoder network
# shp: sharpen operation
# snn: PAWS soft nearest neighbour classifier
# H: cross entropy loss
# ME_max: mean entropy maximization regularization loss
# N: total number of epochs
# use_swa: boolean to use SWA
# use_ema: boolean to use EMA
# swa_epochs: number of epochs before switching on SWA
# gamma: momentum parameter for EMA

g.params = f.params # initialized as copy
g.params.requires_grad = False # remove gradient computations
num_swa = 0
for i in range(N):
    for x in loader:
        x1, x2 = augment(x), augment(x) # augmentations for x
        p1, p2 = snn(f(x1)), snn(f(x2))
        q1, q2 = snn(g(x1)), snn(g(x2))

        loss = H(p1, shp(q2))/2 + H(p2, shp(q1))/2 + ME_max(cat(q1,q2))
        loss.backward()
        optimizer.step()

        if use_swa:
            if i > swa_epochs: # update aggregate
                num_swa += 1
                g.params = (g.params * num_swa + f.params) / (num_swa + 1)
            else:
                g.params = f.params # weights are just copied
        elif use_ema:
            g.params = gamma * g.params + (1-gamma) * f.params # update momentum aggregate
        else:
            g_params = f.params # PAWS baseline
```

---

## D  FURTHER IMPLEMENTATION DETAILS

### D.1  HYPERPARAMETERS FOR VARIOUS THRESHOLD-MEDIATED METHODS

All threshold-mediated methods in this study uses an optimization loss function of the form of Eq. (1), with differences in the hyperparameters $\mu$, $\lambda$, $\tau$, $\rho_t$ and $\alpha$. We use the hyperparameters from the original implementations. For Pseudo-Labels (Lee, 2013), $\mu = 1$, $\tau = 0.95$ and $\rho_{t=0}$ (i.e. hard labels); for UDA (Xie et al., 2019a), $\mu = 7$, $\tau = 0.8$, $\rho_{t=0.4}$ (i.e. soft pseudo-labels sharpened with temperature of 0.4); for FixMatch (Sohn et al., 2020), $\mu = 7$, $\tau = 0.95$, $\rho_{t=0}$ (i.e. hard labels). All methods use $\lambda = 1$. In addition, UDA and FixMatch uses asymmetric transforms for the two legs of sample and pseudo-label prediction, i.e. $\alpha_1$ is a weak transform (based on the standard flip-and-shift augmentation) and $\alpha_2$ is a strong transform (based on RandAugment (Cubuk et al., 2019)). Pseudo-Labels uses symmetric weak transforms for both legs.

In our calibrated versions of all these methods (i.e. "BAM-X"), we maintained the exact same hyperparameter configuration as its corresponding baseline. The only exception is the sharpening temperature of BAM-UDA, which uses $t = 0.9$ instead of $t = 0.4$ in UDA, as we found that calibration enables, and was highly effective with, the use of soft pseudo-labels (see discussion in Section 6).

### D.2  IMPLEMENTATION DETAILS FOR IMAGENET EXPERIMENTS

We maintain the default 64 GPU training configuration recommended by the authors and make slight modifications to the default implementations to fit on our hardware. On our set up (which does not use Nvidia's Apex package due to installations issues), we found training to be unstable on half-precision and had to use full-precision training. In order to fit into memory, we had to decrease the number of images per class from 7 to 6, resulting in a slightly lower baseline performance from the reported for 200 epochs of training (see Table 4 in Assran et al. (2021) for the study on the correlation between the number of images per class and the final accuracy). On all ImageNet experiments on PAWS, we follow the validation and testing procedure of PAWS (Assran et al., 2021) and swept over the same set of hyperparameters during fine-tuning of the linear head. We report the ECE at the final checkpoint.

# E  LONG-TAILED CIFAR-10 AND CIFAR-100

## E.1  DATASET PREPARATION

We create long-tailed versions of CIFAR-10 and CIFAR-100 following the procedure from Cao et al. (2019), i.e. by removing the number of training examples per class from the standard training set with 50,000 samples. We create the class-imbalance unlabeled dataset with an exponential decay where the severity of the imbalance is given by the imbalance ratio $\alpha = \max_i(n_{u,i})/\min_i(n_{u,i}) \in \{10, 100\}$, where $n_{u,i}$ is the number of unlabeled examples for class $i$. The number of samples in the most frequent class is 5,000 for CIFAR-10 and 500 for CIFAR-100. To create the labeled set, we randomly select 10% of samples *from each class*, under the constrain that at least 1 sample for each class is included in the labeled set, i.e. $n_{l,i} = \min(1, 0.1 * n_{u,i})$, where $n_{l,i}$ is the number of labeled examples for class $i$. The total number of labeled and unlabeled examples for the different benchmarks in CIFAR-10-LT and CIFAR-100-LT are summarized in Table 6. The test set remains unchanged, i.e. we use the original (class-balanced) test set of the CIFAR datasets with 10,000 samples.

Table 6: **Dataset statistics for CIFAR-10-LT & CIFAR-100-LT** showing total number of examples in the labeled and unlabeled datasets for each benchmark.

|  | CIFAR-10-LT | | CIFAR-100-LT | |
|---|---|---|---|---|
|  | $\alpha = 10$ | $\alpha = 100$ | $\alpha = 10$ | $\alpha = 100$ |
| Labeled | 2,041 | 1,236 | 1,911 | 1,051 |
| Total | 20,431 | 12,406 | 19,573 | 10,847 |

We used the exact same configuration and hyperparameters as the original (non-long-tailed) CIFAR benchmarks, i.e. the architecture is WideResNet-28-2 for CIFAR-10-LT and WideResNet-28-10 for CIFAR-100-LT. All training hyperparameters used to obtain the results for FM and BAM-UDA in Table 3 also follow the ones from the original CIFAR benchmarks.

## E.2  ADDITIONAL CLASSWISE RESULTS

Fig. 5 shows the test accuracies for the baseline (FM) and ours (BAM-UDA) after separating samples from the test set into three groups — the "Many" group which contains classes with more than 100 samples, the "medium" group which contains classes between 10 and 100 samples and the "few" group which contains classes less than 10 samples, for the benchmark of CIFAR-100-LT with $\alpha = 10$ and 10% labels. We see that BAM-UDA outperforms FM on every group; in addition, the gap between BAM-UDA and FM increases as the number of samples are more scarce, highlighting BAM-UDA's utility in improving accuracy over the baseline in the more difficult classes.

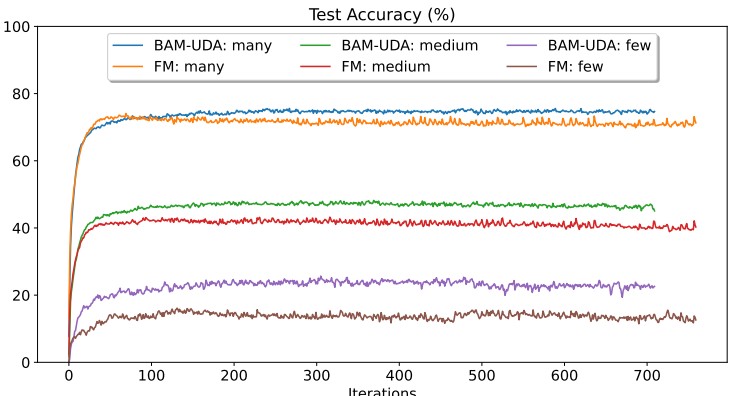

Figure 5: **CIFAR100-LT**. Accuracies for samples based on their their classwise sample frequency, "many" for classes with >100 samples, "medium" for classes between 10-100 samples and "few" for classes between <10 samples. Dataset is CIFAR-100-LT with $\alpha = 10$ and 10% labels.

## F  SEMI-SUPERVISED LEARNING IN PHOTONICS SCIENCE

Semi-supervised learning is highly important and applicable to domains like Science where labeled data is particularly scarce, owing to the need for computationally expensive simulations and labor-intensive laboratory experiments for data collection. To demonstrate the effectiveness of our proposed techniques in the Science domain, we use an example problem in Photonics, adopting the datasets from Loh et al. (2022). The task we studied in this work is a 5-way classification of photonic crystals (PhC) based on their band gap sizes.

PhCs are periodically-structured materials engineered for wide ranging applications by manipulating light waves (Joannopoulos et al., 2008) and an important property of these crystals is the size of their band gap (often, engineers seek to design photonic crystals that host a substantial band gap (Christensen et al., 2020)). We adopt the dataset of PhCs from Loh et al. (2022), which consists of 32,000 PhC samples and their corresponding band gaps which had been pre-computed through numerical simulations (Johnson & Joannopoulos, 2001). Examples of PhCs and an illustration of band gap from the dataset is shown in Fig. 6. We binned all samples in the dataset into 5 classes based on their band gap sizes; since there was a preponderance (about 25,000) of samples without a band gap, we only selected 5,000 of them in order to limit the severity of class imbalance and form a new dataset with just 11,375 samples (see Fig. 6c). Notably the long-tailed distribution of this dataset is characteristic of many problems in science (and other real-world datasets), where samples with the desired properties (larger band gap) are much rarer than trivial samples. From this reduced dataset, we created two PhC benchmarks with 10% labels (PhC-10%) and 1% labels (PhC-1%), where 10% and 1% of samples *from each class* are randomly selected to form the labeled set respectively. The class-balanced test set is fixed with 300 samples per class (total of 1,500 samples), the unlabeled set consists of 9,876 samples and the labeled sets consists of 1,136 samples and 113 samples for PhC-10% and PhC-1% respectively.

For these benchmarks, we used a WideResNet-28-2 architecture, with a single channel for the first CNN layer, and made the following changes upon FixMatch and BAM-UDA. We set $\mu = 1$ and instead of $2^{20}$ iterations, we trained for 300 epochs (where each epoch is defined as iterating through the unlabeled set once). For BAM-UDA, we set $Q = 0.9$ and use a one-minus-cosine warm-up scheduler of 50 epochs to the KL loss coefficient (resulting in a coefficient going from 0 to 1 in 50 epochs). For each benchmark, we swept the learning rate across $\{0.01, 0.001\}$ and select the best model for both the baseline and ours. The standard image augmentations used in vision problems cannot be applied to this problem, since it destroys the scientific integrity of the data (e.g. cropping the PhC input would result in a completely different band gap profile). Instead, we used the augmentations proposed in Loh et al. (2022) (periodic translations, rotations and mirrors) and applied them symmetrically (i.e. there is no distinction of strong and weak augmentations). In addition, we included these augmentations to the labeled samples as well.

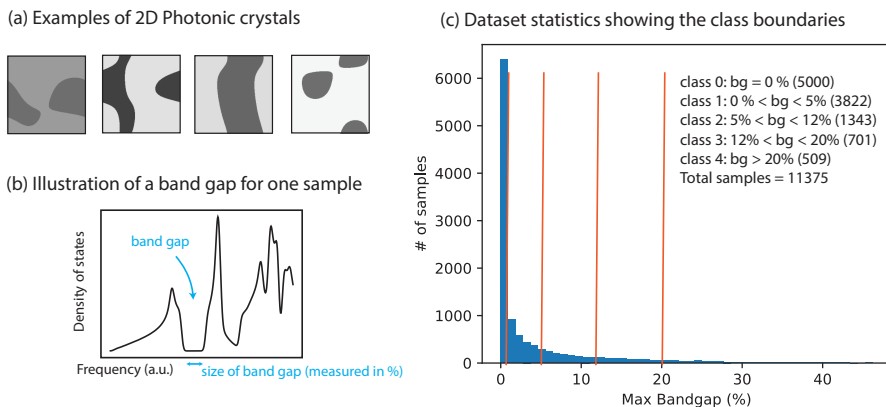

Figure 6: **Photonics dataset.** (a) Examples of 2D periodic photonic crystals (PhC), (b) illustration of the band gap size of a single PhC seen from its density-of-states, a common spectrum of interest for PhCs (Loh et al., 2022), and (c) dataset statistics for the task used in this work where the samples were binned into 5 classes based on their band gap (bg) with class boundaries detailed in the inset. Numbers in parenthesis show the number of samples in each class, showing strong class imbalance.

# G  ABLATION STUDIES

## G.1  WEIGHT AVERAGING IN THRESHOLD-MEDIATED METHODS

For our experiments incorporating EMA during pseudo-labeling for FixMatch, we use the second momentum scheduler defined in Section 5, i.e. a one-minus-cosine scheduler starting at $(1 - x)$ and decreasing to 0, resulting in a $[x, 1]$ range for $\gamma$. While PAWS is effective with high $\gamma$ values (i.e. $x = 0.95$), we found that FixMatch needed much smaller momentum values at the start of training and set $x = 0.25$. Note that in the default implementation of FixMatch, an EMA of constant $\gamma = 0.999$ is also used, however it is used only for *test evaluation* and not during training. In our experiments, we use the EMA weight aggregate *for pseudo-labeling* throughout training and use the same weight aggregate after training for test evaluation. Results are shown in Table 7, where we see that using weight averaging was also effective at improving calibration in FixMatch and can lead to improved test accuracies. However, EMA weight averaging was not so effective when the labels are less scarce and further, it consistently under-performs the Bayesian model averaging approach.

Table 7: **FM+EMA** compared against the baseline FixMatch (FM) as well as the Bayesian model average approach on FixMatch (BAM-FM).

|          | CIFAR-10 | | CIFAR-100 | |
|----------|-----------|-----------|-----------|-----------|
|          | 40 | 250 | 400 | 4000 |
| FM       | 91.7 / 0.078 | 94.2 / 0.051 | 56.4 / 0.366 | 74.2 / 0.183 |
| BAM-FM   | 93.6 / 0.058 | 95.0 / 0.044 | 59.0 / 0.331 | 74.8 / 0.171 |
| FM+EMA   | 93.5 / 0.060 | 94.8 / 0.047 | 56.8 / 0.350 | 74.0 / 0.188 |

The desirability of Bayesian model averaging over EMA weight averaging for threshold-mediated SSL methods could be attributed to replacing the selection metric with better uncertainty estimates given by the BNN layer, which is not available for EMA weight averaging approaches. To further investigate this claim, we perform ablation studies on BAM-UDA to isolate the contribution of averaging predictions from the contribution of replacing the selection metric. Results are shown in Table 8, rows indicated with "BNN no $\sigma^2$" show experiments using a BNN layer in BAM-UDA *only for averaging predictions* while maintaining the original selection metric, i.e. pseudo-labels are accepted if the maximum prediction class probability is greater than $\tau = 0.95$. Comparing the first two rows, indeed we see that by not using the uncertainty estimates from the BNN, we get marginal improvements especially when labels are not so scarce ($72.9 \rightarrow 73.0$ for CIFAR-100-4000 labels),

corroborating our observations with FM+EMA. The difference between the last two rows show the effect of replacing the selection metric and indeed we observe consistent gains across the benchmarks from doing so.

Table 8: **Uncertainty estimate by BNN.** Ablating the importance of uncertainty estimate provided by the variance of BNN predictions. "BNN no $\sigma^2$" indicates that the BNN layer is only used for bayesian model averaging, i.e. predictions are replaced by the posterior predictive but selection metric still follows the baseline, i.e. pseudolabels are accepted if maximum logit value $> 0.95$.

|  | CIFAR-100 | |
| --- | --- | --- |
|  | 400 | 4000 |
| UDA (t=0.4) | 44.0 / 0.491 | 72.9 / 0.185 |
| BAM-UDA, BNN no $\sigma^2$ (t=0.4, $\tau$=0.95) | 48.3 / 0.418 | 73.0 / 0.184 |
| BAM-UDA, BNN no $\sigma^2$ (t=0.9, $\tau$=0.95) | 54.2 / 0.368 | 74.5 / 0.170 |
| BAM-UDA (t=0.9) | 59.7 / 0.327 | 75.3 / 0.167 |

## G.2 ABLATION EXPERIMENTS ON BAYESIAN MODEL AVERAGING

From our main results in Section 6, we found that our Bayesian model averaging approaches were more effective in conjunction with soft pseudo-labels. In Table 9, we show ablation experiments on the temperature of the sharpening operation on the CIFAR-100-400 labels benchmark. Overall, we observe a trend that softer pseudo-labels (i.e. reducing the sharpening of pseudo-labels) led to better calibration and improved test performance. As such, in our experiments we modify upon the original sharpening parameter of UDA and set $t = 0.9$ for BAM-UDA in all our benchmarks.

Table 9: Ablation of sharpening temperature in BAM-UDA. Dataset is CIFAR-100 with 400 labels. Highlighted in cyan is the main configuration used.

| $t$ | Test Accuracy | ECE |
| --- | --- | --- |
| 0.4 | 57.9 | 0.344 |
| 0.8 | 58.1 | 0.340 |
| 0.9 | 59.7 | 0.327 |
| 1.0 | 59.2 | 0.334 |

Bayesian model averaging produces better calibrated predictions through "Bayesian marginalization", i.e. by averaging over multiple predictions instead of using a single prediction of the model. In Table 10, we show ablation studies on the number of weight samples taken from the variational posterior in the BNN layer used for "Bayesian marginalization", i.e. for computing the posterior predictive. We observe an overall trend that increasing the number of weight samples lead to better calibration and final test accuracies, providing direct evidence for the effectiveness in using Bayesian model averaging towards improving calibration. In order to limit the computational overhead arising from taking a large number of weight samples, given that pseudo-labeling happens at every iteration, we limit $M$ to 50 in our study. The computational overhead for $M = 50$ is discussed in Appendix H, where we see that incorporating Bayesian model averaging incurs negligible computational overhead.

Table 10: Ablation of number of weight samples, $M$, taken from the variational posterior in BAM-UDA. Dataset is CIFAR-100 with 400 labels. Highlighted in cyan is the main configuration used.

| $M$ | Test Accuracy | ECE |
| --- | --- | --- |
| 2 | 50.0 | 0.403 |
| 5 | 58.1 | 0.342 |
| 10 | 58.5 | 0.336 |
| 50 | 59.7 | 0.327 |

## G.3 ABLATION EXPERIMENTS ON PAWS-SWA & PAWS-EMA

**Training times before switching on SWA.** Fig. 7 shows the test accuracy and ECE across training for PAWS-SWA and PAWS on the CIFAR-10 benchmark with 4000 labels and explores the effect of varying the time in which SWA was switched on. In particular, we found that switching on SWA later during training is more desirable. Unlike EMA which has a natural curriculum to give more weight to the more recent network parameters in the aggregate, SWA weighs all models in the aggregate equally and thus including the poorly optimized weights from the start of training is undesirable.

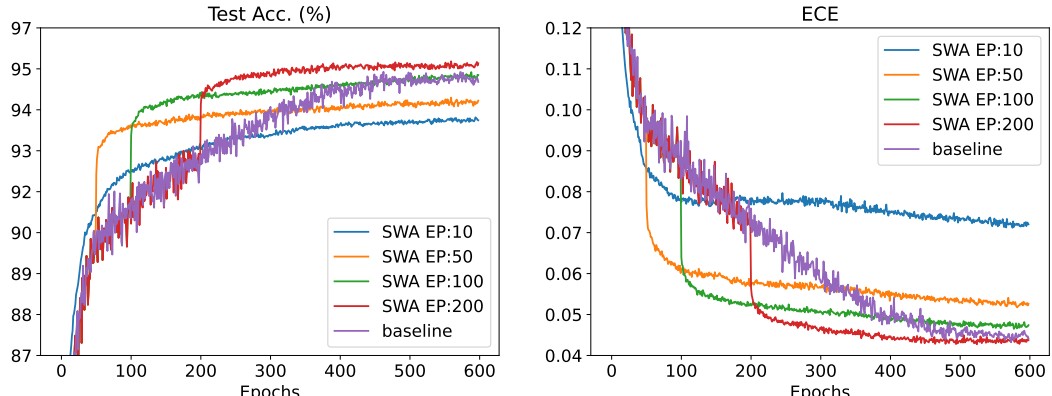

Figure 7: **PAWS-SWA** for different training times before switching on SWA. We experimented with switching on SWA after $\{10, 50, 100, 200\}$ epochs of training (of the baseline). Dataset is CIFAR-10 with 4000 labels.

**Momentum schedules for PAWS-EMA.** Table Table 11 explores different schedules for the $\gamma$ parameter used in PAWS-EMA on the CIFAR-100, 4000 labels benchmark. For brevity, we consider a simple schedule where $\gamma$ is linearly warmed up from 0 to $\gamma_{max}$ in the first $W$ epochs, and maintains at $\gamma_{max}$ thereafter. In Table 11, we varied $W \in \{0, 10, 50\}$ and $\gamma_{max} \in \{0.5, 0.9, 0.99, 0.999\}$. We observe that using a very high $\gamma$ (i.e. 0.999) from the start of training is detrimental to both test performance and ECE, while fixing $\gamma$ to a moderately high value (i.e. 0.9) across training was also less desirable than stepping up $\gamma$ in the first 50 epochs. Notably, these observations highlight distinct differences with the EMA commonly used in dual-network methods of self-supervised learning (Grill et al., 2020b; He et al., 2020) where a very high momentum value (0.996 to 0.999) is usually recommended.

With this insight, in this work, we explored two schedules for $\gamma$: 1) linear warm-up from 0 to 0.996 for 50 epochs and maintaining at 0.996 (highlighted in cyan in Table 11) and 2) using one-minus-cosine schedule starting from 0.05 and decreasing to 0, resulting in a 0.95 to 1.0 range for $\gamma$. These two schedules are visualized in Fig. 8. We use (1) for the CIFAR benchmarks and we found (2) to be more effective on the ImageNet benchmark.

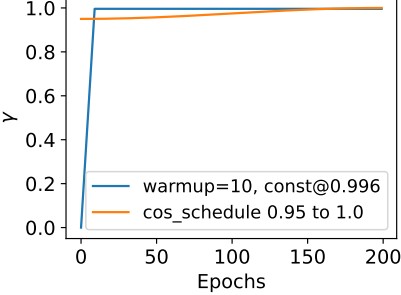

Figure 8: **PAWS-EMA:** Visualization of the momentum schedules used in this work.

Table 11: Ablation on momentum schedule. Dataset is CIFAR-100 with 4000 labels. Highlighted in cyan is the main configuration used for the CIFAR benchmarks.

| warm-up epochs | $\gamma_{max}$ | Test Accuracy | ECE |
|---|---|---|---|
| 0 | 0.5 | 72.1 | 0.215 |
| 0 | 0.9 | 72.4 | 0.218 |
| 0 | 0.99 | 72.0 | 0.208 |
| 0 | 0.999 | 59.2 | 0.266 |
| 10 | 0.99 | 72.6 | 0.198 |
| 10 | 0.999 | 68.5 | 0.214 |
| 50 | 0.99 | 73.2 | 0.193 |
| 50 | 0.999 | 73.3 | 0.200 |
| 50 | 0.996 | 73.6 | 0.189 |

## H    COMPUTATIONAL REQUIREMENTS AND ADDITIONAL COMPUTATIONAL COSTS

All CIFAR-10 and CIFAR-100 experiments in this work were computed using a single Nvidia V100 GPU. ImageNet experiments were computed using 64 Nvidia V100 GPUs. A key aspect of our proposed calibration methods is the requirement of adding minimal computational cost to the baseline approaches. In the following paragraphs we list the additional computational cost (based on wall-clock time on the same hardware) for our proposed methods.

**Computational cost of Bayesian model averaging.**    The main additional computational cost comes from executing several (in our case, 50) forward passes through the weight samples of the BNN layer when deriving predictions for the unlabeled samples. This additional overhead is minimal since only the final layer, which consists of a small fraction of the total network weights, is (approximate) Bayesian. We tested this on the benchmarks of CIFAR-100 and found the BNN versions to take only around $2 - 5\%$ longer in wall-clock time on the same hardware when compared to the baseline, for the same total number of iterations. This justifies the BNN layer as a plug-in calibration approach with low computational overhead.

**Computational cost of SWA & EMA.**    Both SWA and EMA requires storing another network of the same architecture, denoted $g$ in the main text, which maintains the aggregate (exponentially weighted aggregate) of past network weights for SWA (EMA). Rather than using representations from the original backbone, $f$, representations from $g$ are used to derive the better calibrated predictions, and thus the main computational overhead comes from the second forward pass needed per iteration. We timed the methods and found that PAWS+SWA and PAWS+EMA took around 18% longer in wall clock training time (on the same hardware) when compared to the baseline PAWS method for each epoch of training. However, as shown in the main text, PAWS+SWA and PAW+EMA resulted in a significant speed up in convergence, cutting training time by $> 60\%$ and additionally giving better test performances. Hence rather than an overhead, improved calibration in PAWS+SWA and PAWS+EMA in fact reduces the computation cost needed for the original approach.

## I    SOCIETAL IMPACT AND ETHICAL CONSIDERATIONS

Semi-supervised learning is arguably one of the most important deep learning applications, as in real life we often have access to an abundance of unlabeled examples, and only a few labeled datapoints. Our work highlights the importance of calibration in semi-supervised learning methods which could yield benefits for real-world applications in two main ways: 1) a better deep learning model that is more data efficient and 2) a model that is better calibrated and can quantify uncertainty better. The latter is highly important for crucial societal applications such as in healthcare. However, as with any deep learning application, there may be biases accumulated during dataset collection or assimilated during the training process. This issue may be more acute in a semi-supervised setting where the

small fraction of labels may highly misrepresent the ground truth data. This may lead to unfair and unjust model predictions and give rise to ethical concerns when used for societal applications.

