# OpenReview forum: "On the Importance of Calibration in Semi-supervised Learning"
_ICLR.cc/2023/Conference — Submitted to ICLR 2023_

### Official Review · Reviewer_ZkiF · 2022-10-23

**Confidence:** 4
**Correctness:** 4
**Technical Novelty And Significance:** 2
**Empirical Novelty And Significance:** 3
**Recommendation:** 5

**Clarity, Quality, Novelty And Reproducibility:**

**Quality and Clarity**

Good. As stated above, this paper is technically sound and comprehensive with an excellent background introduction. Furthermore its writing is also user-friendly.

**Novelty/Originality**

Seems limited as it is a work brining some already built techniques to a particular scenario. It would be a good application paper but not a paper showing great ML technical depth.

Furthermore the relationship of calibration and accuracy seems not new as well despite that it might be the first-time to state it in SSL. It is a not a rarely known thing in typically supervised learning scenario that calibration at least correlates with accuracy. A (maybe not that good) example is the good (training time) calibration of a simple logistic regression model is just theoretically derived under the assumption that it goes to a perfect solver state (i.e., gradient coming to zero for its bias term).

**Reproducibility**

Good


**Strength And Weaknesses:**

**Strength**

1. The problem this paper considers is important: improving calibration under SSL. Calibration is indeed a very important metric in many real-world industry settings (like computational advertising) while is a comparatively neglected thing in deep learning research, let alone for the specific setting of semi-supervised learning. It's great that the authors could dive deeply into this important problem which might benefit the success of (semi-supervised) deep learning method in real-world scenarios.

2. This paper is stated in a very clear way, and its technical soundness is good. I enjoy reading the whole contents which give readers a good background introduction and "logical flow".

3. The empirical evaluation looks comprehensive and sound.

**Weakness**

1. The technical innovations seems limited: it seems strait-forward to directly adopt the two methods (i.e., bayesian based and weights averaging based) to the scenario of SSL. There is no specific innovations here given no unique challenges (not in terms of theoretical but in terms of algorithmic) when we want to combine the two worlds, that is previous "basic" methods already established and SSL.

**Summary Of The Paper:**

This paper centers around one problem of improving calibration under the scenario of semi-supervised (deep) learning. Towards this end it proposed two methods: the bayesian model averaging based and the weight averaging based. Empirical verifications are conducted on several tasks which demonstrate the improvement both in terms of test set accuracy and the calibration (measured by ECE).

It also points out the correlation between calibration and accuracy as a side outcome.

**Summary Of The Review:**

As stated above, this paper is very well motivated, technically sound and written in a very clear way. Meanwhile its major limitation is on its marginal technical innovation since it is a work connecting the "raw" methods to a specific ML application scenario.

---

> ### Author Response · Authors · 2022-11-09
> **Addressing your concerns**
>
> We thank the reviewer for taking time to review our work. In the following we provide a point-by-point reply to the issues raised.
>
> > 1. The technical innovations seems limited: it seems strait-forward to directly adopt the two methods (i.e., bayesian based and weights averaging based) to the scenario of SSL.
>
> We respectfully disagree that our work lacks technical novelty and kindly point the reviewer to the clarification of our technical novelty in our general response above. In sum, in our work we introduced a new SSL method, BAM-, which presents an entirely new way of generating pseudo-labels using the posterior predictive distribution. We also include a new figure, Figure 1 to clarify the algorithmic novelty. We further contextualize the novelty of BAM- in the context of SSL in a newly added Table 5 in Appendix A, to demonstrate significant algorithmic novelty when compared to prior art.
>
> > There is no specific innovations here given no unique challenges (not in terms of theoretical but in terms of algorithmic) when we want to combine the two worlds, that is previous "basic" methods already established and SSL.
>
> We disagree that “there are no unique challenges” in making BAM- work for SSL. BAM- presents a new way of pseudo-labelling by parameterizing the weights as distributions, which is trained via variational inference and such variational methods have not been explored in SSL to the best of our knowledge. Firstly, it is not directly obvious that BAM-, where we put standard Gaussian priors on the weights can even lead to improvements (particularly since the original authors [1] demonstrate significant degradation of performance when using standard Gaussian priors in the supervised setting). Secondly, many details have gone into the design process to make BAM- work; for example, BAM- replaces the naive selection metric of max logit thresholding with the variance of predictions. This introduces the algorithmic challenge of how to define the threshold since the variance is unbounded unlike the max logit (which we can intuitively impose a threshold close to 1). In our work, we proposed a quantile measure to overcome this challenge.
>
> [1] Weight uncertainty in neural network (https://arxiv.org/pdf/1505.05424.pdf)
>
> > Furthermore the relationship of calibration and accuracy seems not new as well despite that it might be the first-time to state it in SSL. It is a not a rarely known thing in typically supervised learning scenario that calibration at least correlates with accuracy.
>
> While the correlation between accuracy is known in the context of supervised learning, we disagree that we are simply bringing already built techniques from supervised learning into SSL. The solution we proposed in this work (BAM-) is rather unique to SSL since the exact same solution was found to degrade performance in the supervised setting as we pointed out above.

---

> ### Author Response · Authors · 2022-11-16
> **Looking forward to your feedback**
>
> Dear Reviewer ZkiF,
>
> We would be grateful if you can confirm if our response has addressed your concerns and let us know if any issues remain. In the following, we summarize the key points of our response:
>
> - We clarified the novelty of our contribution in our general response above and added new illustrative figures (Fig 1) and a table (Table 5) to exemplify our algorithmic novelty in the paper.
> - We provide justification that BAM- is a rather unique solution to SSL and does not work as effectively in a supervised setting.
>
> We look forward to hearing your feedback!

---

### Official Review · Reviewer_x2Kw · 2022-10-25

**Confidence:** 4
**Correctness:** 3
**Technical Novelty And Significance:** 3
**Empirical Novelty And Significance:** 3
**Recommendation:** 5

**Clarity, Quality, Novelty And Reproducibility:**

This paper presents the idea clearly. The technical novelty is limited. The addressed problem of poor model calibration is a realistic problem. The solutions should be easy to reproduce.


**Strength And Weaknesses:**

Strength:

1. Deep learning model are poorly calibrated. This could affect semi-supervised learning and this work addressed this important problem.

2. Demonstrating a strong correlation between SSL performance and model calibration.

Weakness:

1. The Bayesian network only considers the last layer. In particular, for the pseudo label approach only the classifier head is sampled multiple times. Does this really reveal the uncertainty of a network? This is equivalent to perturbing the decision boundary, the samples that are close to decision boundary are more likely to yield high variance and will be pruned out for self-training. In another words, this can be seen as another way of thresholding.

2. The improvement from FixMatch is very marginal. Does this suggest BAM is another way of doing thresholding.

3. The threshold on Bayesian prediction exploits the distribution of standard deviation on unlabeled data. In contrast, FixMatch does not exploit this information. Thus, it is vital to verify whether the improvement is attributed to a more carefully chosen threshold.

4. EMA parameter update has been studied in existing SSL methods, e.g. mean teacher.

5. It has been reported that confirmation bias is more severe at low label regime. If improving model calibration is effective, it is recommended to evaluate at even lower label regime, e.g. CIFAR-10 @ 40 labeled.


**Summary Of The Paper:**

This work is motivated by the poor model calibration which affects semi-supervised learning. Bayesian network and weight-averaging techniques are adopted to improve model calibration and it has been demonstrated to improve semi-supervised learning efficacy.


**Summary Of The Review:**

Overall, this paper addressed an important problem, e.g. poor model calibration affects SSL performance. However, some components, e.g. using the quantile to estimate threshold, of the proposed method might give unfair advantages over the existing method. The improvement from existing methods is also marginal, thus hard to quantify the significance.

---

> ### Author Response · Authors · 2022-11-09
> **Addressing your concerns**
>
> We thank the reviewer for taking time to review our work. In the following we provide a point-by-point reply to the issues raised.
>
> > 1. The Bayesian network only considers the last layer. In particular, for the pseudo label approach only the classifier head is sampled multiple times. Does this really reveal the uncertainty of a network? This is equivalent to perturbing the decision boundary, the samples that are close to decision boundary are more likely to yield high variance and will be pruned out for self-training. In another words, this can be seen as another way of thresholding.
>
> We thank the reviewer for this interesting insight and fully agree with the reviewer’s intuition about the importance of pruning samples close to decision boundaries. Indeed the reviewer’s intuition also explains why BAM presents a more desirable way of thresholding as compared to using the maximum logit value often used in SSL prior art, since the maximum logit does not imply anything about the samples being near decision boundaries or not. However, we highlight that better thresholding is only one aspect of BAM and does not preclude its role in uncertainty quantification (see the next discussion point).
>
> > 2. The improvement from FixMatch is very marginal. Does this suggest BAM is another way of doing thresholding.
>
> No, BAM has two desirable features: bayesian marginalization (averaging) and thresholding. We demonstrate this through our ablations in Table 8 in Appendix G1, where we used logit-based thresholding similar to FixMatch(FM) (i.e. accept if max prob > 0.95) to replace the variance-based thresholding in BAM and demonstrate we still get significant improvements in accuracy. For convenience, we show the results here for CIFAR-100 @ 400 labels:
>
> Baseline UDA using logit threshold: 44.0% \
> BAM-UDA using logit threshold: 54.2% (using sharpen=0.9) \
> BAM-UDA using variance threshold: 59.7%
>
> We would also like to clarify that while BAM-FM may seem marginal over FM, it is more accurate to compare the improvements of our main method BAM-UDA over FM (as we showed it is important to have soft labels when using BAM-). BAM-UDA achieves non-marginal improvements over FM.
>
> > 3. The threshold on Bayesian prediction exploits the distribution of standard deviation on unlabeled data. In contrast, FixMatch does not exploit this information. Thus, it is vital to verify whether the improvement is attributed to a more carefully chosen threshold.
>
> See the results from the ablation studies in Table 8, Appendix G1, which are reproduced above. The improvement is only partially attributed to a better threshold (and partially attributed to bayesian marginalization).
>
> > 4. EMA parameter update has been studied in existing SSL methods, e.g. mean teacher.
>
> We point the reviewer to our discussion of our novelty and EMA in the general response above. In sum, our main technical novelty is in BAM and we do not claim novelty in the EMA update. The main purpose of EMA is to demonstrate generality in the importance of calibration. We have improved the clarity in our contributions (see modified contribution no. 4) to explicitly acknowledge EMA as a “well-established” method.
>
> > 5. It has been reported that confirmation bias is more severe at low label regime. If improving model calibration is effective, it is recommended to evaluate at even lower label regime, e.g. CIFAR-10 @ 40 labeled.
>
> As the reviewer suggested, we have added an experiment on CIFAR-10 @ 40 labels and obtained good results there as well:
>
> FixMatch (best baseline):  Acc: 92.3 | ECE: 0.067 \
> BAM-UDA (ours):  Acc: 94.7 | ECE: 0.066
>
> Given limited time, we selected the best baseline FixMatch to most critically evaluate BAM-UDA against. We are currently running more experiments and will update our main Table once they complete.
>
> > However, some components, e.g. using the quantile to estimate threshold, of the proposed method might give unfair advantages over the existing method.
>
> The components BAM- uses (e.g. quantile) were exactly introduced in our work to directly tackle the poor calibration in existing SSL methods which rely on a very arbitrary confidence estimate of logit thresholding. We feel it is not fair to call the remedies we introduce as “unfair advantages”.

---

> > ### Comment · Reviewer_x2Kw · 2022-12-09
> > **Final Recommendation**
> >
> > Dear Authors,
> >
> > Thanks very much for providing additional evaluations and explanations. Overall, this paper introduced an interesting problem but some techniques introduced lead me to believe the effectiveness is attributed to better selected thresholds. Therefore, I would like to keep my original rating but it would not disappoint me if this work is eventually accepted.
> >
> > Best
> > Reviewer x2Kw

---

> ### Author Response · Authors · 2022-11-16
> **Looking forward to your feedback**
>
> Dear Reviewer x2Kw,
>
> We would be grateful if you can confirm if our response has addressed your concerns and let us know if any issues remain. In the following, we summarize the key points of our response:
>
> - We provide ablation experiments to demonstrate that BAM- is not simply a better thresholding method, but instead, its effectiveness is derived from both better thresholding and better uncertainty quantification (from bayesian marginalization).
> - We clarified the novelty of our contribution in our general response above and added new illustrative figures (Fig 1) and a table (Table 5) to exemplify our algorithmic novelty in the paper. We also provide clarification and explicit acknowledgements for EMA.
> - We followed your suggestions and added new experiments for the severely low label regime, CIFAR-10 @ 40 labels and showed good results there as well.
>
> We look forward to hearing your feedback!

---

### Official Review · Reviewer_11ZK · 2022-10-25

**Confidence:** 3
**Correctness:** 4
**Technical Novelty And Significance:** 3
**Empirical Novelty And Significance:** 3
**Recommendation:** 5

**Clarity, Quality, Novelty And Reproducibility:**

The paper is well-presented and makes a practical contribution for semi-supervised learning. Reproducibility is possible through the provided code, but more implementation details can be helpful.

**Strength And Weaknesses:**

Strength:

- The paper is well-written and easy to follow. The presented contribution is clearly presented and thus it is compared with the prior work.

- The presented observations can have an impact on a wide range of semi-supervised learning tasks. Moreover, the theoretical result further supports the paper.

- The results are convincing in the three evaluations.

- The paper presents several ablation studies to support why improving the classifier calibration would improve the performance.


Weaknesses:

- It would be interesting to examine more calibration approaches. It is interesting to know if only the presented calibration approach has a positive impact on semi-supervised learning or in general any calibration approach would improve it. Making a more general claim would add significant value to the paper.

- The approach is applied only for classification. Similarly, it could have been applied to the classifier of an object detector.

- Although the paper makes interesting observations, it does not show new elements. For instance, the EMA update is known, as well as the rest parts of the presented methodology. However, the combination of the calibration with semi-supervised learning is a solid contribution.

**Summary Of The Paper:**

The paper proposes to improve the pseudo-label in semi-supervised learning with a well-calibrated model. To that end, it improves the calibration with Bayesian model averaging. It also provides a theoretical result for the benefit of the calibration in semi-supervised learning for the task of classification. The approach is evaluated on CIFAR-10, CIFAR-100 and ImageNet where it clearly demonstrates the advantage of maintaining a calibrated classifier.


**Summary Of The Review:**

The paper shows strong evidence of why the model calibration can improve the performance in semi-supervised learning. It is well-written, with complete related work and a detailed experimental section. Overall, it has potential to be accepted, but there are still some parts to be further worked.


Post-rebuttal comments:

The rebuttal discusses all open questions from the reviews. Regarding my concerns:

- The argumentation that is not possible to compare with another approach does not help the paper. It is necessary to provide more references of comparison for understanding the practical contribution of the paper.

- The lack of novelty is mentioned in another review too. As mentioned, I find the combination a solid contribution but not enough for acceptance. A strong evaluation would add more contribution to the paper (above point).

For these reasons, I will lower my score. The paper needs additional work at the current stage.

---

> ### Author Response · Authors · 2022-11-09
> **Addressing your concerns**
>
> We thank the reviewer for taking time to review our work. In the following we provide a point-by-point reply to the issues raised.
>
> > It would be interesting to examine more calibration approaches. It is interesting to know if only the presented calibration approach has a positive impact on semi-supervised learning or in general any calibration approach would improve it. Making a more general claim would add significant value to the paper.
>
> To the best of our understanding, calibration approaches in literature can materialize in two ways: using Bayesian techniques to create models that are inherently well-calibrated by the virtue of uncertainty quantification or doing post-hoc calibration using a held out validation set. The latter is not ideal in the standard semi-supervised setting since these methods do not assume access to a validation set and given that labels are already scarce. In our work, we mainly focused on the former and operated under a strong constraint that the proposed calibration approach should not stack significant computational overhead (see Appendix H for a discussion on the computational requirements) and thus would preclude any full- or near-full Bayesian approaches as well as deep ensembles [1]. Given these constraints, we demonstrate on three practical approaches (BAM, SWA and EMA) that calibration improves SSL performance using a combination of empirical and theoretical results. We further provide extensive studies on multiple axes: 1) across standard vision benchmarks, 2) across SOTA SSL methods, 3) across calibration methods and 4) across domains (e.g. in science).
>
> [1] Simple and scalable predictive uncertainty estimation using deep ensembles (https://proceedings.neurips.cc/paper/2017/file/9ef2ed4b7fd2c810847ffa5fa85bce38-Paper.pdf)
>
> > The approach is applied only for classification. Similarly, it could have been applied to the classifier of an object detector.
>
> We thank the reviewer for this interesting suggestion. In our work we have mainly focused on the image classification task as it is highly canonical in SSL. We would like to highlight that on top of standard benchmarks commonly studied in SSL works, we have further included experiments in less explored areas such as long-tailed datasets and even on a problem outside of the vision domain, i.e. a real world scientific application. We feel that we have covered sufficient breadth on the applications front. We however acknowledge the reviewer’s suggestion as an interesting future direction.
>
> > Although the paper makes interesting observations, it does not show new elements. For instance, the EMA update is known, as well as the rest parts of the presented methodology. However, the combination of the calibration with semi-supervised learning is a solid contribution.
>
> We thank the reviewer for the positive remarks about our contribution. We however respectfully disagree that our paper “does not show new elements” and kindly point the reviewer to our detailed discussion of our novelty in the general response above. In sum, we are in fact introducing an entirely new SSL algorithm in our work which uses variational inference to learn a posterior distribution which allows pseudo-labels to be generated using the posterior predictive (see our new Figure 1) and further provided theoretical results to motivate our approach. We further show the novelty of BAM- in the context of SSL in a new Table 5 in Appendix A. As the reviewer had also pointed out, the combination of calibration with SSL is a solid contribution and we believe that the community can benefit from such insights.
>
> > The paper is well-presented and makes a practical contribution for semi-supervised learning. Reproducibility is possible through the provided code, but more implementation details can be helpful.
>
> We have included a new illustrative figure, Figure 1 in the main text to better illustrate how BAM- works. We have further included a detailed explanation of the algorithms and their pseudo-code in Appendix C, full implementation details in sections 5 and Appendix D. If the reviewer could also clarify which aspects of our implementation were unclear, we would love to hear and would be happy to supplement these details.

---

> ### Author Response · Authors · 2022-11-16
> **Looking forward to your feedback**
>
> Dear Reviewer 11ZK,
>
> We would be grateful if you can confirm if our response has addressed your concerns and let us know if any issues remain. In the following, we summarize the key points of our response:
>
> - We clarified the novelty of our contribution in our general response above and added new illustrative figures (Fig 1) and a table (Table 5) to exemplify our algorithmic novelty in the paper.
> - We provide further justification for the calibration approaches we proposed or examined in our work.
>
> We look forward to hearing your feedback!

---

### Official Review · Reviewer_bpxv · 2022-11-04

**Confidence:** 3
**Correctness:** 2
**Technical Novelty And Significance:** 2
**Empirical Novelty And Significance:** 2
**Recommendation:** 3

**Clarity, Quality, Novelty And Reproducibility:**

Many of the techniques introduced in the paper have been previously introduced for semi-supervised learning. There are also some issues regarding the paper presentation and claims which are detailed below:

1- The authors introduced ECE metric and they claimed a strong correlation between SSL accuracy and ECE. However, Figure 1 does not show a strong correlation between ECE and accuracy. For example, for FixMatch while the ECE for the threshold of 0.99 is the lowest, the accuracy of this specific case is lower than the threshold of 0.95 with the ECE over 0.18.

2- The paper claims "propose and explore weight averaging approaches" which is the commonly-used teacher-student framework in SSL.

3- Following the introduction of ECE metric, a loss called ELBO based on negative log-likelihood is introduced. However, it is not clear and never discussed how this loss can control ECE.

4- Regarding the experiments the authors "consider the median of 20 checkpoints around the best accuracy checkpoint as the convergence criteria, and report this value as the test accuracy". However, I believe the test accuracies should be reported in the standard format of SSL methods by reporting the variance of the results of different runs with different seeds.

5- It is not clear in the experiments how M weights from the BNN layer are sampled. Did you use drop-out for it?

6- In the abstract, the author claimed up to 15.9% accuracy across different datasets however the improvement is only limited to BAM-UDA on CIFAR-100 with 400 labels.

7- In the experiments, for BAM-PAWS over CIFAR-10 with 250 labels application of SWA or EMA dramatically decreased the accuracy which is unusual. Weight averaging has shown established improvements in the SSL framework. Similarly, for CIFAR-100 with 4000 labels, the results show the same ECE 0.193 while showing different test accuracies.

8- In the experiments, the results should be compared with uncertainty quantification methods.

minor issues

-- In Figure UDA, FixMatch, BAM-UDA across training on CIFAR-100 with 400 labels are compared. However, FixMatch needs to be compared with BAM-FixMatch instead of BAM-UDA.
-- The definition of L_d^l and \delta needs to be presented in Corollary 1.
-- typo: BAyesian Model averaging (BAM)
-- In the supplements, for the proof of Theorem 1 the terms with 1/N_u and 1/N_l cannot be merged to 1/N.

**Strength And Weaknesses:**

Strengths:
The paper is well-written and clear.

Weaknesses:
The main weaknesses rest on the novelty of the introduced techniques.

**Summary Of The Paper:**

The paper leverages approximate Bayesian techniques, such as approximate Bayesian neural networks to quantify the uncertainty and improve the performance on the uncalibrated models. This paper is mainly a summarization of recently introduced techniques in SSL and the
contribution in terms of exploited techniques being commonly used in the SSL framework

**Summary Of The Review:**

The contributions from this paper are not significant and many of the discussed topics and techniques for SSL are well explored in the literature. In the experiments, the variance of different runs with different random seeds needs to be reported.

---

> ### Author Response · Authors · 2022-11-09
> **Addressing your concerns (part 1)**
>
> We thank the reviewer for taking time to review our work. We respectfully disagree that the techniques introduced in this paper have already been introduced for SSL and point the reviewer to our discussion of the novelty in our general response above. In sum, we are in fact introducing an entirely new SSL algorithm in our work which uses variational inference to learn a posterior distribution which allows pseudo-labels to be generated using the posterior predictive and further provided theoretical results to motivate our approach. To the best of our knowledge, such an approach has never been studied in SSL. We further address the issues point-by-point below; we believe we were able to address the main concerns (i.e. novelty, random seeds and some misinterpretation of results) and kindly ask the reviewer to consider increasing their score.
>
> > 1- The authors introduced ECE metric and they claimed a strong correlation between SSL accuracy and ECE. However, Figure 1 does not show a strong correlation between ECE and accuracy. For example, for FixMatch while the ECE for the threshold of 0.99 is the lowest, the accuracy of this specific case is lower than the threshold of 0.95 with the ECE over 0.18.
>
> As the reviewer correctly pointed out, Figure 2 does not show monotonic increase/decrease between accuracy/ECE, however it does demonstrate a strong trend between the two. We compute the Pearson correlation coefficient to be -0.94 (i.e close to -1), verifying strong correlation.
>
> > 2- The paper claims "propose and explore weight averaging approaches" which is the commonly-used teacher-student framework in SSL.
>
> We have updated the paper to replace that line “we propose and explore weight averaging approaches” to “we explored well-established weight averaging approaches”.
>
> > 3- Following the introduction of ECE metric, a loss called ELBO based on negative log-likelihood is introduced. However, it is not clear and never discussed how this loss can control ECE.
>
> The ELBO or evidence lower bound is a lower-bound to the log-marginal likelihood function commonly used in variational inference. BAM uses this loss to do approximate Bayesian inference in the final layer of the model. As Bayesian models are well calibrated, this approach lowers the ECE. We explain this in detail in section 4.1 and Appendix C1. We have now added a line to clarify the link between uncertainty quantification in Bayesian approaches to calibration (ECE).
>
> > 4- Regarding the experiments the authors "consider the median of 20 checkpoints around the best accuracy checkpoint as the convergence criteria, and report this value as the test accuracy". However, I believe the test accuracies should be reported in the standard format of SSL methods by reporting the variance of the results of different runs with different seeds.
>
> We thank the reviewer for this suggestion and fully agree that the variance should be reported. We have updated our main results (Table 1) to include error bars for the most significant results and baselines. Given the limited time, we will continue to update the remaining entries when our experiments complete.
>
> > 5- It is not clear in the experiments how M weights from the BNN layer are sampled. Did you use drop-out for it?
>
> We have added a new figure to illustrate our BAM- method in Figure 1. The weights of the BNN layer are parameterized as distributions instead of point values and thus we can simply sample from these distributions M times to obtain M weights. There is no need and no motivation for using drop-out here. We also refer the reviewer to section 4.1 of the paper for a detailed explanation. The pseudocode is in Algorithm 1 of Appendix C1 and shows the relative simplicity during implementation.
>
> > 6- In the abstract, the author claimed up to 15.9% accuracy across different datasets however the improvement is only limited to BAM-UDA on CIFAR-100 with 400 labels.
>
> As the reviewer correctly pointed out, we do in fact achieve “up to” 15.9% (now 16.2% after averaging over multiple seeds) as we claimed in the abstract. We have updated the abstract to explicitly specify the exact benchmark we achieved this result on.
>
> > 7- In the experiments, for BAM-PAWS over CIFAR-10 with 250 labels application of SWA or EMA dramatically decreased the accuracy which is unusual. Weight averaging has shown established improvements in the SSL framework. Similarly, for CIFAR-100 with 4000 labels, the results show the same ECE 0.193 while showing different test accuracies.
>
> There is a misunderstanding in the reading of our results in Table 1. The lines “+SWA” and “+EMA” are both with respect to the baseline PAWS and not to BAM-PAWS and so the accuracy improvements we show in green would make sense. We have updated Table 1 to instead show “PAWS+SWA” and “PAWS+EMA” to avoid this confusion.
>
> <To be continued in the next thread>

---

> > ### Author Response · Authors · 2022-11-09
> > **Addressing your concerns (part 2)**
> >
> > > 8- In the experiments, the results should be compared with uncertainty quantification methods.
> >
> > Could the reviewer kindly clarify what they mean by a comparison to uncertainty quantification methods? In our work, we are, in fact, using uncertainty quantification (either Bayesian or Frequentist) techniques (BAM or SWA/EMA) to derive better calibrated classifiers and show they improve accuracy when working with limited amounts of data.
> >
> > **Other minor issues:**
> > > -- In Figure UDA, FixMatch, BAM-UDA across training on CIFAR-100 with 400 labels are compared. However, FixMatch needs to be compared with BAM-FixMatch instead of BAM-UDA.
> >
> > We have added the plot of BAM-FM to Figure 3.
> >
> > >  -- The definition of L_d^l and \delta needs to be presented in Corollary 1.
> >
> > $L_d^l$ is simply the generalization error as we stated “is given by..”, we have now added an extra parenthesis to explicitly define this. $\delta$ is already defined in the Corollary as a measure of probability (“with probability at least $1-\delta$ ...”).
> >
> > > -- In the supplements, for the proof of Theorem 1 the terms with 1/N_u and 1/N_l cannot be merged to 1/N.
> >
> > We have added a clarification in the proof of Theorem 1 that $N=N_u + N_l$. Therefore the two terms can be merged since they are Monte-Carlo estimates of the expectation of the NLL loss under the training data distribution.

---

> ### Author Response · Authors · 2022-11-16
> **Looking forward to your feedback**
>
> Dear Reviewer bpxv,
>
> We would be grateful if you can confirm if our response has addressed your concerns and let us know if any issues remain. In the following, we summarize the key points of our response:
>
> - We clarified the novelty of our contribution in our general response above and added new illustrative figures (Fig 1) and a table (Table 5) to exemplify our algorithmic novelty in the paper.
> - We added error bars for different random seeds in our results and obtain slightly greater improvements
> - We addressed some misinterpretation of the results.
>
> We look forward to hearing your feedback!

---

### Author Response · Authors · 2022-11-09
**General Response to all reviewers**

We would like to thank the reviewers for their constructive suggestions, which has allowed us to improve our paper. We believe that we were able to respond, in depth, to all of the concerns and kindly ask the reviewers to consider increasing their scores.

## Novelty
We acknowledge that many reviewers had brought up the issue of novelty and we take responsibility that our initial exposition did not clarify the novel contribution of our work well.

The main novelty in our approach lies in the following two contributions :
1. We proposed a new semi-supervised algorithm (BAM-) which parameterizes the model’s final layer as a distribution over the weights instead of point values. Using this, we propose an entirely new approach of generating pseudo-labels by sampling from the posterior predictive, which by the virtue of quantifying uncertainty leads to better calibrated model and hence significantly improves the performance. We provide theoretical results to formally link the approximate Bayesian approach of BAM to better generalization bound.

2. We provide insight to understanding the empirical success of established techniques in SSL like EMA, i.e. their effectiveness can be understood from an improvement in the calibration of pseudo-labels. To the best of our knowledge, we are the first to empirically demonstrate the link of EMA’s success to the calibration of pseudo-labels.

To remedy the lack of clarity in the original exposition, we have now included a new figure, Figure 1, to provide visual illustration of our new method BAM- and to highlight its algorithmic novelty compared to prior art. BAM- (specifically BAM-UDA) is supported by overall superior performance across most benchmarks in our main results of Table 1.

To further highlight the novelty of BAM in the context of SSL, we added a new table, Table 5 in Appendix A (reproduced below), which shows the progress of SSL prior art. SSL methods have thus far been differing combinations of techniques focused along two axes: augmentation and/or post-processing of pseudo-labels (e.g. sharpening, thresholding). Comparatively, BAM presents significant differences algorithmically using a variational inference approach and obtaining pseudo-labels from the posterior predictive; to the best of our knowledge a variational approach has never been explored in SSL. We think this constitutes sufficient display of novelty and further believe the community can benefit from our work – both from BAM’s algorithmic novelty in SSL as well as the insights we provide regarding the importance of calibrating pseudo-labels.

| SSL method      | Augmentation   | Pseudo-label                     | Selection Metric           |
|-----------------|----------------|----------------------------------|----------------------------|
| Mean teacher    | Weak           | EMA                              | -                          |
| UDA             | Weak & Strong  | Sharpen                          | Logit threshold            |
| MixMatch        | Weak           | Averaging aug + sharpen          | -                          |
| FixMatch        | Weak & Strong  | Hard labels                      | Logit threshold            |
| FlexMatch       | Weak & Strong  | Hard labels                      | Class-wise logit threshold |
| **BAM- (Ours)** | Weak & Strong  | **Posterior sampling** + sharpen | **Variance threshold**     |

## EMA
Almost all of the reviewers pointed out that one of the methods we explored in our work, EMA, is a well-established technique in SSL in the mean-teacher framework. We would like to clarify that we do not claim method novelty over this existing technique and have previously cited them. The primary reason for including EMA (as well as the proposed SWA which has not been studied in the context of SSL) was to provide further evidence that improving calibration improves model performance goes beyond a Bayesian layer approach (BAM-). As reviewer 11ZK also pointed out, it is important to examine more calibration approaches in order to make such a general claim. A natural approach to show generality is via ensemble methods that have been highly successful in uncertainty quantification, of which weight-ensembling methods like EMA and SWA qualify a part of. To better clarify that we do not claim method novelty over EMA, we have now separated our contribution on calibration approaches into two, with contribution no. 4 explicitly stating that EMA is well-established in SSL:
> “We further explored weight averaging techniques, one of which (i.e. EMA) being well-established in SSL and show…”

We have also incorporated all the suggestions from the reviewers and updated our paper, with changes indicated in blue.

---

### Decision · Program_Chairs · 2023-01-20

**Decision:**

Reject

**Justification For Why Not Higher Score:**

- Lack of comparisons to other calibration techniques
- Lack of specific novel techniques to address the identified problem

**Justification For Why Not Lower Score:**

N/A

**Metareview: Summary, Strengths And Weaknesses:**


This paper proposes a new semi-supervised learning algorithm based on the observation that classifier accuracy and calibration are correlated in the context of SSL. Reviewers generally appreciated the importance of studying calibration in the SSL context, the clear presentation, and promising results in the evaluations. However, they had concerns about the technical innovation of the method given that its elements are well-studied in the SSL context, as well as the lack of comparison to other techniques that can also improve calibration. The authors provided responses to these concerns, but did not include additional comparisons to other techniques; reviewers did not feel their concerns were adequately addressed by the response. Overall, reviewers unanimously lean towards rejection due to the novelty concerns and lack of comparison to other techniques.

The AC appreciates the perspective that the combination of techniques used to address the identified calibration issue in SSL is novel and interesting, but also agrees that other calibration techniques should be included in the experiments. This could help to further motivate the design of the proposed method and to provide more evidence that calibration is indeed the mechanism behind the improved performance; these were also points of concern for reviewers. The empirical evidence used to support the correlation between calibration and accuracy could also be further strengthened by evaluating over more settings (datasets, models, label budgets). The authors are encouraged to strengthen the submission accordingly for submission to a future venue.